

# High variability of export fluxes along the North Atlantic GEOTRACES section GA01: Particulate organic carbon export deduced from the $^{234}$Th method

Nolwenn Lemaitre[1,2,3,] Frédéric Planchon[2], Hélène Planquette[2], Frank Dehairs[3], Debany Fonseca-Batista[3,4], Arnout Roukaerts[3], Florian Deman[3], Yi Tang[5], Clarisse Mariez[2], Géraldine Sarthou[2]

[1]Department of Earth Sciences, Institute of Geochemistry and Petrology, ETH-Zürich, Zürich, Switzerland
[2]Laboratoire des Sciences de l'Environnement Marin (LEMAR), UMR 6539, IUEM, Technopôle Brest Iroise, 29280 Plouzané, France
[3]Vrije Universiteit Brussel, Analytical, Environmental and Geo-Chemistry, Earth System Sciences research group, Brussels, Belgium
[4]Oceanography Department, Dalhousie University, Halifax, Nova Scotia, Canada
[5]Department of Earth and Environmental Sciences, the Graduate Center, City University of New York, New York, USA

*Correspondence to*: Nolwenn Lemaitre (nolwenn.lemaitre@erdw.ethz.ch)

**Abstract.** In this study, we report Particulate Organic Carbon (POC) export fluxes estimated using the $^{234}$Th-based approach in different biogeochemical basins of the North Atlantic, as part of the GEOTRACES GA01 expedition (GEOVIDE, May-June 2014). Surface POC export fluxes were deduced by combining export fluxes of $^{234}$Th with the POC to $^{234}$Th ratio of sinking particles at the depth of export. Particles were collected in two size classes (>53 µm and 1-53 µm) using *in-situ* pumps and the large size fraction was considered as representative of sinking material. Surface POC export fluxes revealed latitudinal variations between provinces ranging from 1.4 mmol C m$^{-2}$ d$^{-1}$ in the Irminger basin where the bloom was close to its maximum peak, to 12 mmol C m$^{-2}$ d$^{-1}$ near the Iberian Margin where the bloom had already declined. In addition to the bloom staging, the variations of POC export fluxes were also related to the phytoplankton community structure. In line with previous studies, the presence of coccolithophorids and diatoms appeared to increase the POC export flux while stations dominated by pico-phytoplankton cells, such as cyanobacteria, were characterized by lower fluxes. The surface POC export fluxes were then compared to *in-situ* and satellite primary production (PP) in order to assess the export efficiency. This ratio strongly varied regionally and was generally low ($\leq$ 14 %), except at two stations located near the Iberian margin (35%) and within the Labrador basin (38%), which were characterized by unusual low *in-situ* PP. We thus conclude that the North Atlantic during this period was not as efficient in exporting carbon from the surface, as described in recent studies. Finally, we estimated the flux of POC exported 100 m below the surface export depth in order to investigate the transfer efficiency along the section. This parameter



was also highly regional-dependent but the lowest attenuation of the POC flux was observed at stations where
coccolithophorids dominated.

## 1. Introduction

Through the sinking of particulate biogenic material, the biological carbon pump (BCP) plays a major role on the
sequestration of carbon-rich particles in the ocean interior. The North Atlantic harbors one of the most productive
spring phytoplankton bloom of the world's ocean (Esaias et al., 1986; Longhurst, 2010), generating an important pulse
of biogenic sinking particles (Buesseler et al., 1992; Honjo and Manganini, 1993; Le Moigne et al., 2013), which
accounts up to 18 % of the global BCP (Sanders et al., 2014). As a consequence, the North Atlantic has been identified
as an efficient ocean to export carbon to depth (Buesseler and Boyd, 2009; Herndl and Reinthaler, 2013).
The North Atlantic consists of several provinces (sensu Longhurst, 1995) characterized by distinct biogeochemical
and physical characteristics. For example, the low nutrient availabilities in the North Atlantic subtropical gyre limits
the biomass development (Moore et al., 2008), dominated by pico-phytoplankton such as cyanobacteria (Zehr and
Ward, 2002). Northward, in the Irminger and Labrador basins, the phytoplankton growth is strongly seasonally light-
limited (Riley, 1957) and the key parameter for alleviating these limitations is the progressive shoaling of the mixed
layer. There, diatoms dominate the phytoplankton bloom until the exhaustion of the silicic acid stock (Martin et al.,
2011). Then, an intense bloom of coccolithophorids develops (Poulton et al., 2010; Raitsos et al., 2006).There, diatoms
dominate the phytoplankton bloom until the exhaustion of the silicic acid stock (Martin et al., 2011). Then, an intense
bloom of coccolithophorids develops (Poulton et al., 2010; Raitsos et al., 2006). Between these basins, the west
European and Icelandic basins are like a transition zone where nutrients and/or light can limit the primary production
(Henson et al., 2009).  In summer, at the end of the bloom, these basins can be iron-limited, and become "High Nutrient
Low Chlorophyll" regions (Blain et al., 2004; Moore et al., 2006; Sanders et al., 2005). The North Atlantic is thus a
heterogeneous basin in terms of limitations and phytoplankton communities. Ecosystem structure is thought to play an
important role on the fraction of particulate organic carbon (POC) which is exported from the surface ocean. Indeed,
Guidi et al. (2009) suggested that phytoplankton composition explained 68% of the variance in POC flux at 400 m.
High export efficiencies are reported in productive regions where diatoms dominate, but the exported material is
relatively labile and prone to remineralisation leading to low deep export flux and transfer efficiencies. Conversely, in
oligotrophic regions, where diatoms are largely absent, primary production is low and mostly regenerated.
Consequently, export efficiencies are low but the eventual exported material is likely to be refractory resulting in high
transfer efficiencies (Henson et al., 2012; Lam et al., 2011; Lima et al., 2014; Marsay et al., 2015). Phytoplankton size
structure has also been shown to be an important factor in controlling the POC export fluxes. Guidi et al. (2015)
highlighted that the exported POC was more refractory and the remineralisation depth was greater when the fraction
of micro-phytoplankton decreased or the fraction of pico-phytoplankton increased.
According to the impact of these biogeochemical factors on the POC export, the efficiency of the North Atlantic to
transfer POC to the deep ocean can be questioned. In this context, we investigated the derived-POC export fluxes using
the Thorium-234 ($^{234}$Th) approach in the different basins of the North Atlantic. $^{234}$Th is a highly particle reactive



element, with a short half-life (24.1 days), which is widely used to explore particle export over short time events such
as phytoplankton blooms. A deficit of $^{234}$Th with respect to its radioactive parent $^{238}$U (conservative in seawater) is
usually observed in the upper water column where particles sink. In subsurface, excess of $^{234}$Th compared to $^{238}$U can
be observed pointing to a remineralisation of $^{234}$Th-bearing particles (Savoye et al., 2004). A $^{234}$Th flux can be
quantified and then converted into a POC flux by using the POC:$^{234}$Th ratio of sinking particles at the depth of export
(Buesseler et al., 2006).
In this paper, we first explore the basin-wide variations of the $^{234}$Th fluxes in order to define different export and
attenuation regimes in the North Atlantic. The POC export fluxes are then discussed with regards to the stage and
intensity of the bloom and the phytoplankton structure. Finally, using *in-situ* primary production, satellite primary
production and deeper POC export fluxes, we investigate the export and transfer efficiencies in the North Atlantic.
**2.  Methods**
For the purpose of this work, 11 stations were investigated along the GEOTRACES section GA01 (GEOVIDE cruise,
15 May to 30 June 2014; R/V Pourquoi Pas?). The studied area crossed different basins: the Iberian basin, the west
European basin, the Icelandic basin, the Irminger basin and the Labrador basin (Fig.1).
**2.1.  Description of the regional basins**
The Iberian basin (Stations 1 and 13) was characterized by oligotrophic conditions, with $NO_3^-$ and $Si(OH)_4$
concentrations under 1 μmol L$^{-1}$ in the upper 40 m, despite the proximity of the Iberian margin, where sits a natural
upwelling (Costa Goela et al., 2016; Zúñiga et al., 2016; http://marine.copernicus.eu/), that potentially fuels the area
with nutrient-rich deep waters. Dissolved iron (dFe) concentrations were non-limiting, with concentrations in surface
waters varying between 0.22 and 1.0 nmol L$^{-1}$ (Tonnard et al., 2018; this issue). In this basin, the total chlorophyll-*a*
(Chl-*a*) in the upper 200 m was the lowest averaging 0.26 mg m$^{-3}$ and nano-phytoplanktonic species dominated but
with a mixed proportion of micro-, nano- and pico-phytoplanktonic species. The highest proportion of pico-
phytoplanktonic species was observed at Station 13 (34% of the total Chl-*a*; Tonnard et al., in prep.) with cyanobacteria
contributing for 12% of the total Chl-*a*.
The west European basin (Stations 21 and 26) was influenced by the North Atlantic Current (NAC): the southernmost
sub-branch evolving in a cyclonic eddy at Station 21 and the sub-arctic front (SAF) at Station 26. This front separates
cold and fresh waters from the subpolar region and the warm and salty waters from the subtropical region (Zunino et
al., 2017; this issue). Both stations were characterized by low surface $Si(OH)_4$ concentrations (≤ 1 μmol L$^{-1}$), moderate
$NO_3^-$ (≥ 1 μmol L$^{-1}$) and dFe concentrations (0.17 and 0.18 nmol L$^{-1}$ in the upper 20 m). The total Chl-*a* concentrations
were higher in this basin reaching 0.45 mg m$^{-3}$ (n=2). Micro-phytoplanctonic species dominated, contributing for more
than 45% of total Chl-*a*. The combined contribution of diatoms, dinoflagellates and haptophytes (including
coccolitophorids) was about 71%, with a higher contribution of diatoms (41%) compared to the two other taxa (4 and
26%, respectively; Tonnard et al., in prep.).



Within the Icelandic basin, Stations 32 and 38 were respectively influenced by the NAC northern branch and the
Eastern Reykjanes Ridge Current (ERRC; Zunino et al., 2017; this issue). The surface waters had low Si(OH)$_4$
concentrations (1 µmol L$^{-1}$), relatively high dFe concentrations (> 0.45 nmol L$^{-1}$; Tonnard et al., 2018; this issue) and
high NO$_3^-$ concentrations (> 6 µmol L$^{-1}$). Concentrations of total Chl-$a$ reached 0.62 mg m$^{-3}$ on average in the upper
200 m of Station 32 and 0.44 mg m$^{-3}$ at Station 38, a value close to the one observed in the west European basin. Nano-
phytoplanktonic species dominated, up to 81% of the total Chl-$a$ at Station 38, and this was associated to the highest
proportion of the haptophytes (56 and 55% at Stations 32 and 38; Tonnard et al., in prep.).
In the Irminger basin, Station 44 was located in the Irminger gyre while Station 51, located close to Greenland, was
influenced by the Eastern Greenland Current (EGC) guided by the continental slope (Zunino et al., 2017; this issue).
This basin was characterized by high surface Si(OH)$_4$ and NO$_3^-$ concentrations (> 6 µmol L$^{-1}$) and moderate to high
dFe concentrations (0.24-1.3 nmol L$^{-1}$; Tonnard et al., 2018; this issue). The highest total Chl-$a$ concentrations were
measured in this basin, averaging 0.98 mg m$^{-3}$ (n=2). Micro-phytoplanctonic species, and more specifically diatoms,
clearly dominated with a contribution of about 55 and 77% of the total Chl-$a$ at Stations 44 and 51, respectively
(Tonnard et al., in prep.).
The Labrador basin (Stations 64, 69 and 77) was characterized by the subduction of the Labrador Seawater (LSW)
which was particularly intense (1700 m-deep convection) during the winter 2013-2014 (Kieke and Yashayaev, 2015).
Stations 64 and 77 were also influenced by the Western Greenland Current (WGC) and the Labrador Current (LC),
respectively. Macronutrients in the surface waters of the Labrador basin were characterized by a north to south gradient,
with high NO$_3^-$ and Si(OH)$_4$ concentrations at Station 64 (≥ 4.5 µmol L$^{-1}$), decreasing gradually to the south with low
concentrations at Station 77 (~1 µmol L$^{-1}$ and < 1 µmol L$^{-1}$, respectively). Moderate dFe concentrations were observed
in this area (between 0.23 and 0.30 nmol L$^{-1}$ in the upper 20 m; Tonnard et al., 2018; this issue). As the Irminger basin,
micro-phytoplankton species (diatoms) dominated with a contribution averaging 57% of the total Chl-$a$ (Tonnard et
al., in prep.). However, the total Chl-$a$ concentrations were low (down to 0.25 mg m$^{-3}$), especially at Stations 64 and

123   69.

## 2.2. Total $^{234}$Th and $^{238}$U

Total $^{234}$Th activities were determined from 4 L unfiltered seawater samples collected with Niskin bottles and stored
in polypropylene Nalgene bottles. Usually, 17 or 18 depths were sampled between the surface and 1000-1500 m, except
at Stations 26 and 77 where only 9 and 15 depths were sampled, respectively (Table S1). In addition, deep samples
(n=15; between 1000 and 3500 m) were taken for the calibration of low level beta counting (van der Loeff et al., 2006).
Deep water samples are generally considered to be in secular equilibrium regarding the $^{234}$Th-$^{238}$U pair. Seawater
samples were processed following the method developed by Pike et al. (2005). Briefly, samples were acidified at pH
2 using concentrated HNO$_3$ (suprapur grade, Merck), spiked with 1 mL of $^{230}$Th yield monitor to estimate the $^{234}$Th
recovery after the sample processing. After 12 hours of equilibration, pH was increased to 8.5 using concentrated
NH$_4$OH (suprapur grade, Merck). One hundred micro-liters of KMnO$_4$ and MnCl$_2$ (analytical grade, Merck) were
added to form a manganese oxide precipitate and, after 12 hours of equilibration, samples were filtered on quartz-
microfiber discs (QMA, Sartorius, 1 µm nominal porosity, 25 mm diameter). On board, filters were dried at 50 °C





overnight, mounted on nylon holders, covered with Mylar and aluminum foil and $^{234}$Th activity counted using low
level beta counters (RISØ, Denmark). Beta activity counting was continued until a relative standard deviation (RSD)
≤ 2% was reached. At the home-laboratory, residual beta activity was measured for each sample after a delay of six
$^{234}$Th half-lives (~6 months) and these residual counts were subtracted from the gross counts obtained on-board. All
samples were then processed for Th recovery using $^{229}$Th as a second yield tracer. Briefly, filters were dismounted
from the nylon holders and transferred to clean 30 mL teflon vials (Savillex). All samples were spiked with 50 µL of
$^{229}$Th, dissolved in a 10 mL mix of 8M HNO$_3$/1M H$_2$O$_2$ (suprapur grade, Merck), heated overnight at 60 °C and filtered
through Acrodisc® syringe filters (Pall, Nylon membrane, nominal porosity=0.2 µm, diameter=25 mm). Two
milliliters of the filtrate were pre-concentrated by evaporation and the residue diluted in 6 mL of 1.4 M HNO$_3$ (suprapur
grade, Merck). $^{230}$Th and $^{229}$Th concentrations were measured by SF-ICP-MS (Element 2, Thermo Scientific) in low
resolution mode. Each sample was analyzed 3 times and the precision of the $^{230}$Th:$^{229}$Th ratios averaged 1.2% (RSD),
which is within the range indicated by Pike et al. (2005). The total $^{234}$Th recovery, involving all the steps described
above, was 91 ± 14 % (n=200). Uncertainty on total $^{234}$Th activity was estimated using error propagation and accounts
between 0.04 and 0.10 dpm L$^{-1}$.
The $^{238}$U activity was deduced from salinity using the Equation 1, given by Owens et al. (2011):
$$^{238}U = 0.0786 \times S - 0.315 \qquad (1)$$

where $^{238}U$ is the $^{238}$U activity in dpm L$^{-1}$ and $S$ is the salinity in psu.

### 2.3. Particulate $^{234}$Th and POC sampling and analysis

Suspended particles were collected using *in-situ* large-volume filtration (100-1600 L) systems (Challenger Oceanics
and McLane pumps; ISP hereafter for "*in-situ* pumps") through paired 142 mm-diameter filters: a 53 µm mesh nylon
screen (SEFAR-PETEX®; polyester) and a 1 µm pore size quartz-microfiber filter (QMA, Sartorius), respectively.
Two size fractions of particles were thus collected: the small size fraction (referred to as SSF hereafter, 1-53 µm) and
the large size fraction (referred to as LSF hereafter, >53 µm). Filters were cleaned prior to the cruise as follows: PETEX
screens were soaked in 0.6M HCl, (Normapur, Merck) rinsed with Milli-Q water, dried at ambient temperature in a
laminar flow hood and stored in clean plastic bags; QMA filters were pre-combusted at 450 °C during 4 h and stored
in aluminum foils until use. *In-situ* pumps were deployed on a stainless steel cable between 15 and 800 m and the
pumping time was approximately 2-3 hours (Table S2).
After collection, filters were processed on board. The 142 mm PETEX screen was cut into quarters using a clean
scalpel and two quarters were processed in this study. Large particles collected on the screen were rinsed off using
0.45 µm filtered seawater and re-filtered under a laminar flow hood on a silver filter (SterliTech, porosity=0.45 µm,
diameter=25 mm) for the first quarter of the PETEX screen and on a GF/F filter (Whatman®, porosity=0.7 µm,
diameter=25 mm) for the second quarter.
The QMA filters were sub-sampled with a perspex punch of 25 mm diameter.
Then, silver, GF/F and QMA filters were dried overnight (50 °C) and prepared for beta counting (see Section 2.2).
After counting the residual beta activity (~ 6 months later), samples were prepared for POC, PN analyses along with





their $\delta^{13}$C and $\delta^{15}$N isotopic compositions (here we present only POC data). In brief, filters were dismounted from filter
holders and fumed with HCl vapor overnight inside a glass desiccator to remove the carbonate phase. After a drying
step at 50 °C, samples were packed in precombusted (450 °C overnight) silver cups and analyzed with an elemental
analyzer – isotope ratio mass spectrometer (EA-IRMS, Delta V Plus, Thermo Scientific). Acetanilide standards were
used for the calibration. The detection limits and C blanks were respectively 0.63 and 0.80 µmol for Ag filters (n=11)
and were 0.49 and 1.52 µmol for QMA filters (n=13).
**2.4. Export fluxes of $^{234}$Th**
Thorium-234 activity in surface waters can be described using a simple mass balance equation (Savoye et al., 2006),
which accounts for production from $^{238}$U decay, $^{234}$Th decay, sinking flux and transport as follow:
$$\frac{dA_{Th}}{dt} = \lambda A_U - \lambda A_{Th} - P + V \qquad (2)$$

where $A_{Th}$ is the activity of total $^{234}$Th in dpm L$^{-1}$; $A_U$ is the salinity-derived activity of $^{238}$U in dpm L$^{-1}$, $\lambda$ is the $^{234}$Th
decay constant (0.0288 d$^{-1}$); $P$ is the net removal of $^{234}$Th on sinking particles in dpm L$^{-1}$ d$^{-1}$; $V$ is the sum of the
advective and diffusive fluxes in dpm L$^{-1}$ d$^{-1}$.
Assuming steady state (constant total $^{234}$Th activity with time) and neglecting the physical term V (Buesseler et al.,
1992), the net export flux of particulate $^{234}$Th can be determined using the following equation:
$$P_z = \lambda \int_0^z (A_U - A_{Th}) dz \qquad (3)$$

where $P_z$ is the integrated flux of $^{234}$Th from the surface to the depth z in dpm m$^{-2}$ d$^{-1}$. Equation 3 has been solved for
z representing the depth Eq at the base of the deficit zone (where $^{234}$Th activity is back to secular equilibrium with
$^{238}$U) as well as for z representing the base of the Primary Production Zone (PPZ), i.e. the depth where *in-situ*
fluorescence was only 10% of the surface value (see Section 3.1). The validity of the assumptions used for solving
Equation 3 is discussed in Sections 4.1 and 4.2.
In Section 4.2, we attempt to calculate the $^{234}$Th fluxes at the Eq depth by using a non-steady state (NSS) model (Savoye
et al., 2006) that can be described as follows:
$$P_z = \lambda \left[ \frac{A_U (1 - e^{-\lambda \Delta t}) + A_{Th1} e^{-\lambda \Delta t} - A_{Th2}}{1 - e^{-\lambda \Delta t}} \right] \qquad (4)$$

where $\Delta t$ is the time interval between two visits of a single station; $A_{Th1}$ and $A_{Th2}$ are the $^{234}$Th activities at the first and
second visits, respectively.
In order to account for possible $^{234}$Th excess relative to $^{238}$U below Eq, we evaluated the $^{234}$Th flux at the depth
corresponding to Eq + 100 m, and compared this flux with the $^{234}$Th flux at Eq (Black et al., 2017). The difference
R100 is expressed as follows:
$$R100 = P_{Eq} - P_{Eq+100} \qquad (5)$$



where $R100$ is the flux reduction in dpm m$^{-2}$ d$^{-1}$.
**2.5. Scavenging fluxes of $^{234}$Th**
Using the data for particulate $^{234}$Th activities next to total $^{234}$Th activities, it is possible to describe the $^{234}$Th activity
with a 2-box model (Coale and Bruland, 1985). This model accounts for the partitioning of $^{234}$Th between the
particulate and dissolved phase. The mass balance equation for dissolved $^{234}$Th can be written as follows:
$$\frac{dA_{Thd}}{dt} = \lambda A_U - \lambda A_{Thd} - J + V_d \qquad (6)$$

where $A_{Thd}$ is the activity of dissolved $^{234}$Th in dpm L$^{-1}$; and $J$ is the net removal flux from the dissolved to the particulate
form (scavenging flux) in dpm L$^{-1}$ d$^{-1}$. Here, $^{238}$U is considered as a dissolved specie which produces $^{234}$Th in the
dissolved phase. The second mass balance equation describes the particulate $^{234}$Th pool as follows:
$$\frac{dA_{Thp}}{dt} = J - \lambda A_{Thp} - P + V_p \qquad (7)$$

where $A_{Thp}$ is the activity of particulate $^{234}$Th (in dpm L$^{-1}$); the scavenging flux $J$ described above becomes here the
source term; $P$ is the net removal flux of particulate $^{234}$Th with sinking particles and already described with the one
box model (Eq. 2 and 3). Using again the steady state assumption (constant activities for both particulate and dissolved
$^{234}$Th) and ignoring the physical terms ($V_d$ and $V_p$), Equation 5 becomes:
$$J_z = \lambda \int_0^z (A_U - A_{Thd})dz \qquad (8)$$

where $J_z$ in dpm m$^{-2}$ d$^{-1}$ is the net integrated flux of dissolved $^{234}$Th to the depth z. In our case, the calculation was
performed at the Eq depth for comparison with the $^{234}$Th export flux.
In a similar way, Equation 7 is simplified to:
$$J_z = \lambda \int_0^z A_{Thp}dz + P_z \qquad (9)$$

Under these conditions, the net flux of scavenging J (source term) is defined by two output terms, the export of
particulate $^{234}$Th ($P_z$) due to sinking particles and the sorption of dissolved $^{234}$Th onto non-sinking suspended particles.
**2.6. *In-situ* primary production**
In order to determine the *in-situ* daily PP, stable isotope incubation techniques (H$^{13}$CO$_3^-$) were conducted using
seawater collected in the euphotic zone based on Photosynthetically Active Radiation (PAR) profiles as described
elsewhere (Fonseca-Batista et al., 2018). Briefly, at each station, seawater was sampled from 3 to 6 depths (from 54 to
0.2% of surface PAR) and incubated on deck with a H$^{13}$CO$_3^-$ enriched substrate. All on-board incubations were
sampled at the initial state and after 24h of experiment, seawater was then filtered through microglass fiber filters
(MGF, 0.7 µm porosity, Sartorius). At the home-laboratory, POC concentrations and isotopic composition were
analyzed by EA-IRMS and uptake rates were deduced following the Hama et al. (1983) method. Daily PP was then



estimated for each station by integrating the uptake rates as a function of depth from the surface down to 0.2% of
surface PAR. Note that the sampling to determine the *in-situ* PP at Station 51 occurred 24h after the sampling of the
total [234]Th, particulate [234]Th and POC.

### 2.7. Satellite primary production

In addition to the daily *in-situ* PP described above, PP was obtained from satellite data products available from the
Ocean Productivity website at Oregon State University (http://www.science.oregonstate.edu/ocean.productivity/) with
a 9 km spatial resolution and 8-day temporal resolution obtained from MODIS and SeaWiFS satellites. Three different
models can be used to obtain satellite-derived PP: the standard Vertically Generalized Production Model (VGPM;
(Behrenfeld and Falkowski, 1997), the Eppley-VGPM (Eppley, 1972) and the Carbon-Based Production Model
(CbPM; (Behrenfeld et al., 2005; Westberry et al., 2008). In this study, we present the PP data derived with the VGPM
model, since its output fitted best the *in-situ* PP. Furthermore, Puigcorbé et al. (2017) have shown that the PP data from
the CbPM deviated from *in-situ*, especially in subpolar areas, probably due to the presence of coccolitophorids and
large diatoms that increase disproportionately the backscattering due to their shells and frustules.
PP data were averaged over a $5 \times 5$ pixel box centered on the different sampling sites, corresponding to a surface area
of 2025 km² (45 km $\times$ 45 km). The PP was averaged for the week (8 days), the month (32 days) and the whole
productive period prior to the sampling date.

## 3. Results

### 3.1. Depth distribution and spatial variability of the [234]Th/[238]U disequilibria

The complete dataset of total [234]Th, [238]U activities and the corresponding [234]Th/[238]U ratios are presented in Table S1
and Figure 2 shows the depth profiles of total [234]Th and [238]U activities. A deficit of [234]Th relative to [238]U ratio indicates
a loss of [234]Th due to the export by particles (Buesseler et al., 1992; Cochran and Masqué, 2003). Conversely, excess
of [234]Th relative to [238]U implies [234]Th accumulation, which can be related to particle degradation, releasing [234]Th in the
dissolved phase (Waples et al., 2006). Along the transect, total [234]Th activities ranged between 1.23 and 2.90 dpm L$^{-1}$,
while [238]U activities ranged from 2.19 to 2.53 dpm L$^{-1}$.
At all stations, deficits of [234]Th relative to [238]U were observed in the upper 100 m. The lowest [234]Th/[238]U ratios were
measured in the upper 40 m, ranging from 0.50 (Station 38) to 0.90 (Station 44). Generally, the lowest ratios were
observed in the west European and Icelandic basins (median $0.61 \pm 0.12$, n=4), while in the other basins, the median
surface [234]Th/[238]U ratios was $0.74 \pm 0.06$ (n=8). Moreover, at Stations 21, 26 and 32 located within the west European
and Icelandic basins, ratios below 0.8 deepened further than the other stations ($91 \pm 14$ m, n=3, compared to $33 \pm 16$
m, n=8). Total [234]Th activities increased progressively with depth and were back to equilibrium with [238]U at different
depths between stations: Eq reached $100 \pm 10$ m (n=2) in the Iberian basin, increased to $128 \pm 51$ m (n=4) in the west
European and Icelandic basins and finally decreased to $68 \pm 27$ m in the Irminger and Labrador basins (n=5).
This Eq depth matched relatively well with the base of the PPZ depth, as only 16 m of difference was observed between
both depths in average along the transect and with the biggest difference (~60 m) at Stations 1, 32 and 51 (Fig. 2). The





PPZ depth, defined as the depth at which the fluorescence reaches 10% of its maximum value (Marra et al., 2014), was
used in different studies as the integration depth for $^{234}$Th deficits (Owens et al., 2014; Roca-Marti et al., 2016b) but
the good correspondence between Eq and PPZ confirms that the Eq depth is appropriate to calculate export fluxes.
Below Eq, significant excesses of $^{234}$Th relative to $^{238}$U (i.e., $^{234}$Th/$^{238}$U ratio >1.1) were only observed at Stations 1
(800 m), 13 (140 m), 21 (300 m) and 77 (400 and 700 m). Occasionally a small but significant $^{234}$Th deficit was also
observed at depths deeper than Eq. This is the case for Station 44 at 140 m and 800 m and at Station 51 between 400
and 700 m (Fig. 2).

### 271    3.2. Export and scavenging fluxes of $^{234}$Th

Steady state (SS) $^{234}$Th export fluxes, integrated at the Eq and PPZ depths ranged respectively from 321 to 2282 dpm
m$^{-2}$ d$^{-1}$; and from 321 to 1723 dpm m$^{-2}$ d$^{-1}$ (Table 1). Similar fluxes were found at both integration depths with
differences smaller than 12%, except at Stations 32 and 51 where fluxes at Eq were 36 and 46% greater than those at
the base of the PPZ. Considering that there can be export (or remineralisation) below or above the PPZ depth, in the
following, only the export fluxes at the Eq depth are discussed as they represent the fully-integrated depletion of $^{234}$Th
in the upper waters and thus the maximal export.
The highest $^{234}$Th export fluxes at Eq using the SS model were observed in the west European and Icelandic basins,
reaching 2282 ± 119 dpm m$^{-2}$ d$^{-1}$ at Station 32, while the lowest flux was determined in the Irminger basin (321 ± 66
dpm m$^{-2}$ d$^{-1}$ at Station 44; Table 1).
Using the SS model, we also determined the $^{234}$Th scavenging fluxes at the Eq depth (Equations 8 and 9), along the
transect. These fluxes ranged from 1495 to 3917 dpm m$^{-2}$ d$^{-1}$ at Stations 38 and 21, respectively (Table 1). In general,
the highest $^{234}$Th scavenging fluxes were observed in the west European basin and at Stations 13 and 32 in the Iberian
and Icelandic basins, respectively (median value: 3294 ± 548 dpm m$^{-2}$ d$^{-1}$, n=4). The lowest fluxes were determined in
the Labrador basin and at Stations 1 and 38 in the Iberian and Icelandic basins respectively (1495 ± 176 dpm m$^{-2}$ d$^{-1}$,
n=5).

### 287    3.3. Particulate $^{234}$Th and POC distributions

Particulate $^{234}$Th activities and POC concentrations for the small size fraction (SSF; 1-53 µm) and the large size fraction
(LSF; >53 µm) are presented in Table S2.
LSF particles were collected on silver and GF/F filters (see Section 2.2), and POC concentrations and $^{234}$Th activities
were determined on both filter types. The POC concentrations and $^{234}$Th activities compared well between both filter
types, indicating a relatively good repeatability in sampling ($^{234}$Th$_{GFF}$ = 0.63 × $^{234}$Th$_{silver}$ + 0.01 with r$^2$=0.88, p-
value<0.01 and n=58; and POC$_{GFF}$= 0.86 POC$_{silver}$ + 0.08 with r$^2$=0.90, p-value<0.01 and n=58; Fig. S1). Yet,
concentrations from GF/F filters are systematically lower than the ones from Ag filters, most likely because of the
different pore size filter (0.7 µm for GF/F filter vs 0.45 µm for Ag filter).
High POC concentrations and $^{234}$Th activities were observed in the upper 100 m, ranging respectively from 0.42 to 17
µmol C L$^{-1}$ and from 0.02 to 1.2 dpm $^{234}$Th L$^{-1}$ at Stations 64 and 44 for the SSF and from 0.16 to 4.0 µmol C L$^{-1}$ and
from 0.01 to 0.61 dpm $^{234}$Th L$^{-1}$ at Stations 38 and 44 for the LSF. At all stations, both POC concentrations and $^{234}$Th



activities decreased rapidly in the subsurface to remain essentially constant below 200 m. In general, the lowest POC
concentrations and particulate $^{234}$Th activities were determined in the Iberian basin. Moderate concentrations and
activities were measured in the west European, Icelandic and Labrador basins, except at Station 77 where POC
concentrations and $^{234}$Th activities were higher in surface reaching 11 µmol C L$^{-1}$ and 0.45 dpm $^{234}$Th L$^{-1}$ for the SSF,
and, 3.0 µmol C L$^{-1}$ and 0.20 dpm $^{234}$Th L$^{-1}$ for the LSF. The highest concentrations and activities were measured in
the Irminger basin.
Along the transect, POC contents and $^{234}$Th activities were predominantly carried by the SSF, with POC and $^{234}$Th in
the LSF accounting only for 13 and 12% (median values; n=56) of total POC and $^{234}$Th, respectively. This feature was
accentuated in the Iberian basin, especially at Station 13, where $^{234}$Th in the LSF averaged 6.6 ± 1.3% (median ± 1
s.d.; n=5) of the total particle associated activity and in the Icelandic basin where the LSF represented only 9.2 ± 8.1
% (n=10) of the total POC. The highest proportion of POC and $^{234}$Th in the LSF were observed in surface waters of
the west European basin (reaching respectively 51 and 47%), the Irminger basin (reaching respectively 39 and 56%)
and the Labrador basin (reaching respectively 42 and 51%).
Large variations were also observed along the transect when comparing the fractions of the whole particulate $^{234}$Th
(sum of the LSF and SSF), accounting from 9% (Station 1) to 94% (Station 44) of the total $^{234}$Th activity. The median
value was 26% (n=11) but four stations were characterized by different partitioning compared to the general trend.
Stations 1, 64 and 69 were characterized by a low particulate $^{234}$Th activity accounting for 9, 10 and 15% of the total
$^{234}$Th activity in agreement with the low POC concentrations observed at these stations. Conversely, Station 44 was
characterized by the highest fraction of particulate $^{234}$Th (94%), reflecting an important particle concentration in surface
waters.
**3.4. POC:$^{234}$Th ratios in particles**
Profiles of POC:$^{234}$Th ratios for the SSF and LSF are shown in Figure 3. POC:$^{234}$Th ratios spanned two orders of
magnitude, ranging between 0.51 (Station 1, 800 m) to 53.7 (Station 32, 30 m) µmol dpm$^{-1}$ in the SSF and from 1.05
(Station 21, 400 m) to 30.6 (Station 1, 30 m) µmol dpm$^{-1}$ in the LSF. The highest and most variable ratios were
determined in the upper water column (~30 m) with values ranging from 4.73 µmol dpm$^{-1}$ at Station 13 to 53.7 µmol
dpm$^{-1}$ at Station 32 for the SSF, and from 5.6 µmol dpm$^{-1}$ at Station 38 to 30.6 µmol dpm$^{-1}$ at Station 1 for the LSF.
Then, the ratios decreased with depth, in both size fractions, down to 100 m. Below 100 m, the ratios remained
relatively constant in both size fractions with median values of 1.8 ± 1.1 µmol dpm$^{-1}$ in the Iberian basin (n=13; Stations
1 and 13), 3.0 ± 1.3 µmol dpm$^{-1}$ in the west European and Icelandic basins (n=24; Stations 21, 26, 32 and 38), 3.7 ±
0.9 µmol dpm$^{-1}$ in the Irminger basin (n=10; Stations 44 and 51) and 7.0 ± 3.8 µmol dpm$^{-1}$ in the Labrador basin (n=16;
Stations 64, 69 and 77). The decrease of the POC:$^{234}$Th ratio with depth illustrated the preferential degradation of
carbon relative to $^{234}$Th.
Because the POC to $^{234}$Th ratio has to be determined at the export depth for the conversion of $^{234}$Th flux into POC
export flux (Buesseler et al., 2006), the POC:$^{234}$Th ratios in the LSF and SSF were estimated at this specific depth (Eq
in the present study) using the power law interpolation of the measured ratios. The highest POC:$^{234}$Th ratios at Eq in
the SSF and the LSF were determined in the Labrador basin reaching 16.8 and 13.7 µmol dpm$^{-1}$, respectively, at Station



69. At most stations, the POC:$^{234}$Th ratios at Eq were comparable for both size fractions with differences between the
LSF and SSF smaller than a factor 1.7 (Table S2). The highest differences were determined at Station 1 with the
POC:$^{234}$Th ratios for the LSF being 1.7 fold higher than the one of the SSF, and at Stations 13 and 44 where the
POC:$^{234}$Th ratios for the SSF being 1.5 and 1.7 fold higher than those of the LSF.

### 3.5.  POC export fluxes

We estimated the POC export fluxes by multiplying the $^{234}$Th export flux with the POC:$^{234}$Th ratio, both determined
at the Eq depth. POC fluxes were determined by using the POC:$^{234}$Th ratios of the LSF (>53 µm) and the SSF (1-53
µm; Table 2) in order to compare both estimations.
Except at Stations 1, 26 and 64, the POC fluxes were between 1.1 to 1.5 folds higher when using the SSF ratio.
However, when considering the uncertainties, POC fluxes determined with the POC:$^{234}$Th ratios in SSF and LSF were
not significantly different. Moreover, as we did not have the possibility to compare the ratios with those from sediment
traps, we cannot affirm that the small particles participated to the export. As large and rapidly sinking particles usually
drive most of the export (Lampitt et al., 2001; Villa-Alfageme et al., 2016), most of the studies dedicated to POC
export fluxes in the North Atlantic used the POC:$^{234}$Th ratios from the LSF(Ceballos-romero et al., 2016; Le Moigne
et al., 2013; Moran et al., 2003; Owens et al., 2014; Sanders et al., 2010; Thomalla et al., 2008). Therefore, in the
following, we only discuss the POC fluxes determined with the POC:$^{234}$Th ratios from the LSF (Table 3).
The POC export fluxes at Eq using the LSF ranged from $1.4 \pm 0.5$ mmol m$^{-2}$ d$^{-1}$ at Station 44 to $12 \pm 22$ mmol.m$^{-2}$.d$^{-1}$
at Station 1 and the median was $6.1 \pm 3.3$ mmol m$^{-2}$ d$^{-1}$ (n=11). Besides Station 1 where the POC flux reached $12 \pm 22$
mmol m$^{-2}$ d$^{-1}$, two main open-ocean areas were characterized by high POC export fluxes: 1) the west European and
Icelandic basins, in particular Stations 26 and 32 where POC export fluxes reached 7.9 and 8.3 mmol m$^{-2}$ d$^{-1}$
respectively and 2) the Labrador Sea basin and in particular Station 69 where POC export flux reached $10 \pm 1$ mmol
m$^{-2}$ d$^{-1}$.

### 3.6.  *In-situ* and satellite primary production

*In-situ* PP obtained along the GEOVIDE cruise and discussed in this study are presented and argued in more details
for the subtropical area (Stations 1, 13 and 21) in Fonseca-Batista et al. (2018; this issue). Across the North Atlantic,
the PP varied by a factor of 6, ranging from $27 \pm 5$ at Station 69 to $174 \pm 19$ mmol C m$^{-2}$ d$^{-1}$ at Station 26 (Table 3).
Low PP were determined in the Iberian basin, with one of the lowest values measured at Station 1 (33 mmol C m$^{-2}$ d$^{-1}$
) and a moderate PP at Station 13 (79 mmol C m$^{-2}$ d$^{-1}$; Fonseca-Batista et al., 2018; this issue). The west European
basin was highly productive with PP reaching 135 and 174 mmol C m$^{-2}$ d$^{-1}$ at Stations 21 and 26, respectively.
Similarly, the Station 32, within the Icelandic basin was highly productive with a PP reaching $105 \pm 11$ mmol C m$^{-2}$ d$^{-1}$
$^{1}$ but Station 38 was characterized by a lower PP ($68 \pm 7$ mmol C m$^{-2}$ d$^{-1}$). Within the subpolar area, the PP was high
in the Irminger basin, ranging from $137 \pm 2$ to $166 \pm 32$ mmol C m$^{-2}$ d$^{-1}$ at Stations 44 and 51, respectively, but the PP
was lower in the Labrador basin, ranging from $27 \pm 5$ to $80 \pm 21$ mmol C m$^{-2}$ d$^{-1}$ at Stations 69 and 77, respectively.
In addition to incubation data, we looked at the annual record of satellite-derived PP in order to document the recent
trend in the biological production before the cruise. 8-days averaged PP data for the year 2014 are shown in Figure 4.




The intensity and duration of the productive period was highly variable between basins. Most of the stations were
sampled during the spring bloom period yet at different stages, except Stations 1 and 13 within the Iberian basin, which
were sampled 10 to 12 weeks after the start of the bloom. At these stations, PP increased very early in the year (early
to mid-March) and collapsed rapidly (end of March to mid-April) probably due to the setup of oligotrophic conditions
(Fonseca-Batista et al., 2018). Northward, the stations in the west European basin were the most productive with the
highest PP peak observed at Station 21 (403 mmol m$^{-2}$ d$^{-1}$), 13 days before the sampling. At Station 26 close to the
SAF, the sampling took place during a secondary PP increase. Further north, in the Icelandic and the Irminger basins
the spring bloom period started in May. At sampling time, PP was still increasing at Station 32 while the Stations 44
and 51 as well as the stations of the Labrador basin (64, 69 and 77) were sampled two to three weeks after the bloom
peak.
Using the 8-day average data, PP was estimated for the preceding month (32 days) and the whole productive period
prior to the sampling date in order to account for different timescales in PP and to compare with export fluxes estimates
(Table 3 and Fig. 5). Comparable values (differences smaller than a factor 1.5) were obtained at Stations 13, 21, 32,
38, 44, 64 and 77 between *in-situ* and satellite-derived PP data (8-day average). At the other stations, the *in-situ* data
were larger, up to 2.3 folds (Station 26), or lower, down to 4 folds (Station 69), compared to the 8-day average satellite
data.

## 386    4. Discussion

In the following section, we first discuss the potential impact of the physics and non-steady state conditions on the
$^{234}$Th export flux estimations, prior to defining different $^{234}$Th export and attenuation regimes along the GEOVIDE
transect. In line with the $^{234}$Th regional variability, the POC export fluxes are discussed with regards to the
biogeochemical characteristics of the different basins and compared with published studies in the North Atlantic.
Finally, we examine carbon export and transfer efficiencies along the transect.

### 392    4.1. $^{234}$Th export fluxes under the potential influence of physical conditions

The GEOVIDE section sampled a diversity of dynamic regimes (Zunino et al., 2017; see Section 2.1), including
continental margins affected by strong zonal surface currents (LC, WGC and EGC; Mercier et al., 2015; Reverdin et
al., 2003), a local and seasonal upwelling (close to the Iberian Margin), as well as a deep convection zone in the
Labrador Sea. In such conditions, the Equation 3, which assumes the physical components (lateral and vertical
advective and diffusive fluxes) as negligible, may not always be appropriate (Savoye et al., 2006). Whenever possible,
we explore quantitatively or qualitatively, the potential errors introduced in our calculation.
Lateral processes associated to high velocities currents and intense mesoscale activity are known to affect the $^{234}$Th
distribution (Benitez-Nelson et al., 2000; Resplandy et al., 2012; Roca-Marti et al., 2016b; Savoye et al., 2006). In our
case, this may concern several stations located at or close to margins such as Stations 51 and 64 subject to the powerful
East and West Greenland Currents on the Greenland Margin, Station 77 with the LC on the Newfoundland Margin and
Station 1 with the Portugal Current on the Iberian Margin (Fig. 1). However, the impact of the lateral advection cannot
be quantified from our dataset, as the required horizontal gradients of $^{234}$Th cannot be resolved at a sufficient resolution.





As an alternative, we can compare the stations close to each other, as for instance Stations 44 and 51, both located in
the Irminger Basin where surface currents are strong. The relatively high variability of the $^{234}$Th fluxes (321 and 922
dpm m$^{-2}$ d$^{-1}$, respectively) found at these two stations may indicate a potential influence of lateral advection. The higher
export flux at Station 51 may suggest an input of $^{234}$Th depleted waters originating from the Arctic and/or the Greenland
shelf. However, Arctic (Cai et al., 2010; Roca-Marti et al., 2016a) and Greenland shelf waters (Station 53, see Table
S1) reveal very limited depletions of $^{234}$Th relative to $^{238}$U. Thus, the $^{234}$Th deficit at Station 51 reasonably seems to be
essentially driven by vertical rather than horizontal processes.
The impact of physics concerns also the open ocean sites, such as stations within the west European and Icelandic
basins (Stations 26 and 32) that are subjected to mesoscale activity. An inverse modeling study carried out in that
region, at the Porcupine Abyssal Plain site, suggests that the vertical transport of $^{234}$Th associated with small-scale
structures could represent up to 20% of the estimated vertical export flux (Resplandy et al., 2012). This error is larger
than our analytical uncertainty and should be kept in mind when considering the export flux data in this area.
The vertical advection can also impact the distribution of $^{234}$Th. In upwelling systems, this contribution has been shown
to be important (Buesseler, 1998; Buesseler et al., 1995). Near the Portuguese coasts in the Iberian margin, the intensity
of the upwelling is seasonally dependent (Costa Goela et al., 2016; Zúñiga et al., 2016) and was rather inactive at the
time of the GEOVIDE cruise (http://marine.copernicus.eu/). Therefore, the input of $^{234}$Th-rich deep waters to the
surface is likely to be limited, as already observed in the northern Iberian margin in early summer (Hall et al., 2000).
Downwelling systems, such as the intense convection that occurred in the Labrador basin during the winter prior to
our sampling (Kieke and Yashayaev, 2015), are also prone to impact the $^{234}$Th distribution. However, a strong vertical
advection would homogenize the $^{234}$Th activities in the water column, which is not the case during our study (Fig. 2).
Indeed, the greatest mixed layer depth along the GEOVIDE transect reached 40 m and significant $^{234}$Th deficits relative
to $^{238}$U were observed in surface waters compared to deeper depths. Therefore, the influence of vertical advection on
$^{234}$Th export fluxes was neglected.
Finally, the contribution of the vertical molecular diffusion was estimated using the vertical gradients of total $^{234}$Th
activity in upper waters and a Kz value ranging between $10^{-4}$ and $10^{-5}$ m$^2$ s$^{-1}$, as observed in the upper 1000 m between
Portugal and Greenland along the OVIDE transect (Ferron et al., 2014). The highest vertical diffusive flux was
determined at Station 69 and reached 181 dpm m$^{-2}$ d$^{-1}$, which is in the range of the $^{234}$Th flux uncertainties. Therefore,
the impact of the vertical diffusion has not been further considered.
In conclusion, although considered to have limited or no impact on the measured $^{234}$Th export fluxes, physical
processes should be kept in mind when interpreting these export fluxes.
**4.2. Accounting for non-steady state conditions**
As the cruise sampling scheme did not allow to collect samples through a time series, it was necessary to assume steady
state conditions (i.e., no variation of $^{234}$Th activity with time). However, as documented in previous studies in the west
European and Icelandic basins (Buesseler et al., 1992; Martin et al., 2011), this assumption was invalid and large
variations of $^{234}$Th activity were observed at a time scale of one to three weeks along with the onset of the seasonal
biological productivity. As a consequence, the SS model was shown to poorly describe the magnitude of the $^{234}$Th



export flux as it underestimated fluxes by up to a factor 3 compared to the non-steady state (NSS) model (Buesseler et al., 1992; Martin et al., 2011). Indeed, large changes in satellite-derived PP have been observed during the weeks preceding the sampling (Fig. 4, Section 3.5). Most of the stations were sampled during the most productive period of the year except Stations 1 and 13 sampled in post bloom conditions.

In order to evaluate the potential error introduced by the SS approach, we have attempted to apply a NSS model. Without time series data, the calculation should not be performed sensu stricto (Buesseler et al., 1992; Savoye et al., 2006) but we chose to set the initial conditions for each station, as done by Rutgers van der Loeff et al. (2011) in the South Atlantic (Eq. 4).

Satellite-derived PP data were used to estimate the starting date of the bloom, defined by a PP increase of 30 % above the winter value. $^{234}$Th was assumed to be in equilibrium with $^{238}$U at this time point and the time interval ($\Delta t$) for the calculations stretched from the bloom start until the sampling date. All physical terms were considered negligible. The highest NSS $^{234}$Th fluxes were determined in the west European and Icelandic basins, reaching $3540 \pm 113$ dpm m$^{-2}$ d$^{-1}$ at Station 32, while the lowest flux was determined in the Irminger basin ($516 \pm 90$ dpm m$^{-2}$ d$^{-1}$ at Station 44; Table 1). The NSS $^{234}$Th fluxes were all exceeding or equalling those deduced using the SS model because only a decreasing trend of surface $^{234}$Th activity (i.e., an increasing deficit) was considered in the NSS model. Also, because the initial $^{234}$Th activity is the same for all stations, the differences between NSS and SS fluxes are essentially driven by $\Delta t$. For stations sampled shortly after the start of the bloom such as in the Irminger, Icelandic and Labrador basins ($\Delta t$ ranges from 23 to 43 days), the fluxes predicted by the NSS model are from 1.4 to 2.1 folds higher relative to the SS ones. In the west European and Iberian basins, this difference is reduced (NSS fluxes are from 1.1 to 1.3 folds higher) due to the greater $\Delta t$ (from 48 to 78 days).

As a conclusion, the SS export fluxes may have underestimated $^{234}$Th export fluxes at some stations by a maximum factor of 2 in the Icelandic basin. Yet, we need to keep in mind that this NSS approach has limitations by assuming the equilibrium between $^{234}$Th and $^{238}$U at the bloom start and by considering only an increasing deficit during $\Delta t$.

### 4.3. Surface export regimes of $^{234}$Th

In addition to the export flux (P), we also used the measured partitioning between the particulate and the dissolved phase to estimate the scavenging flux of $^{234}$Th (J). As described in Section 2.4, the scavenging flux accounts for the total removal of $^{234}$Th from the dissolved phase and thus represents the sum of two contributions: the $^{234}$Th sorption flux onto suspended non-sinking particles and the $^{234}$Th export flux via sinking particles. The comparison between the export flux (P) and scavenging flux (J) in terms of P/J ratio (export ratio) offers a valuable metric for estimating the export efficiency of $^{234}$Th. A low P/J ratio indicates that the removal of dissolved $^{234}$Th is controlled by sorption onto suspended particles rather than export. Conversely, a high P/J ratio indicates that the $^{234}$Th is preferentially exported rather than adsorbed and is thus efficiently removed from the upper waters.

Along the GEOVIDE section, the $^{234}$Th export ratios (P/J) varied strongly (Fig. 6), highlighting variable export regimes of $^{234}$Th across the North Atlantic. The most striking feature is the very low value determined in the Irminger basin (as low as 0.2 at Station 44) suggesting that export of $^{234}$Th is particularly inefficient at this location. The retention of $^{234}$Th in the surface layer probably reflects an accumulation phase of biomass in this area. For the other stations, the export



ratio is much higher, ranging from ~0.5 (corresponding to a balanced situation between P and J fluxes) to up to >0.75
indicating an efficient export of $^{234}$Th by sinking particles. Variations can also be observed within the same basin. For
instance, in the Iberian and Labrador basins, Stations 1 and 64 close to the Iberian and Greenland margins respectively,
display much higher export ratios compared to off-shore stations (Stations 13 and 69, respectively). This reflects
different particle dynamics and the more efficient export of $^{234}$Th at these margin stations is possibly related to the
presence of numerous lithogenic particles (Gourain et al., 2018; Lemaitre et al., in prep.), scavenging the $^{234}$Th. The
scavenging of $^{234}$Th onto non-organic particles has already been observed in the North Atlantic, in particular in benthic
nepheloid layers (Owens et al., 2014).
### 4.4. $^{234}$Th export flux attenuation in the upper mesopelagic zone
Excess of $^{234}$Th relative to $^{238}$U is indicative of particle break-up and remineralisation by heterotrophic bacteria and/or
zooplankton (Benitez-Nelson et al., 2001; Black et al., 2017; Buesseler et al., 2008; Cai et al., 2010; Maiti et al., 2010;
Owens et al., 2014; Planchon et al., 2013; Savoye et al., 2004; Usbeck et al., 2002). To estimate the intensity of this
remineralisation just below the upper waters, export flux calculations were extended 100 m below the Eq depth. Note
that conditions of $^{234}$Th excess below Eq yield fluxes integrated until Eq+100 m which are smaller than fluxes
integrated over Eq. As reported in Table 1, the reduction of $^{234}$Th flux (R100, see Section 2.3) was observed only at a
limited number of stations. Evidence for shallow remineralisation (R100 values above uncertainties) can be found in
the Iberian basin (Station 13, R100=410 ± 218 dpm m$^{-2}$ d$^{-1}$), the west European basin (Station 21, R100=360 ± 255
dpm m$^{-2}$ d$^{-1}$) and the Labrador basin (Station 69, R100=401 ± 159 dpm m$^{-2}$ d$^{-1}$ and Station 77, R100=252 ± 165 dpm
m$^{-2}$ d$^{-1}$). Recently, Black et al. (2017) determined R100 values in the southeastern tropical Pacific which are of the
same order of magnitude, averaging 400 ± 200 dpm m$^{-2}$ d$^{-1}$ but with values up to 1200 dpm m$^{-2}$ d$^{-1}$. Relative to the
surface export flux, the flux reductions represented a decrease of 30, 20, 50 and 40% at Station 13, 21, 69 and 77,
respectively. The significant flux attenuation at Stations 13 and 21 likely reflects an important bacterial activity,
reinforced in warm waters (>13°C in the upper 100 m; Iversen and Ploug, 2013; Marsay et al., 2015; Rivkin and
Legendre, 2001). In the Labrador Sea, the particulate biogenic Ba$_{xs}$ proxy also revealed evidence of enhanced
mesopelagic remineralisation, especially at Station 69 (Lemaitre et al., 2018).
For other stations located in the Irminger or Icelandic basins, no apparent decrease in flux was detected suggesting that
shallow remineralisation was probably less intense.
### 4.5. Spatial trends in POC export fluxes
In the North Atlantic, the intensity and the stage of the bloom, as well as the phytoplankton composition significantly
vary spatially. As the $^{234}$Th proxy integrates the activity deficits over a timescale of several weeks preceding the
sampling and as the phytoplankton size structure and composition are known to exert a control on the magnitude of
the POC export flux (Allredge and Jackson, 1995; Boyd et al., 1999; Guidi et al., 2009), it appears essential to compare
the spatial variations of these parameters in order to better understand the spatial variability of POC export. In the
following, the POC export fluxes are discussed according to the different biogeochemical regions sampled during the
survey.



*The Iberian basin*

One of the lowest POC export flux was determined within the Iberian basin, at Station 13 ($2.2 \pm 0.3$ mmol m$^{-2}$ d$^{-1}$) where the PP remained rather low along the season (Fig. 4) and where the highest abundance of pico-phytoplankton was observed (Tonnard et al., in prep.). These conditions are typical of the subtropical and oligotrophic waters (Dortch and Packard, 1989). Villa-Alfageme et al. (2016) highlighted that small cells are usually slow-sinking particles that can be easily remineralised in shallow waters. Their small sinking velocity (<100 m d$^{-1}$) allows time for bacteria and zooplankton to degrade these particles, thus reducing the export flux. In the same area, a low flux was also measured later in the season (October) by Owens et al. (2014), confirming the lower carbon export fluxes in this oligotrophic area. Still in the Iberian basin, the highest POC export flux (albeit the strong associated error) was determined at Station 1 ($12 \pm 22$ mmol m$^{-2}$ d$^{-1}$). As Station 13, Station 1 was sampled after the bloom period and was characterized by low nutrient concentrations but, conversely to Station 13, Station 1 was characterized by a mixed proportion of micro-, nano- and pico-phytoplankton. Moreover, this station was also influenced by lithogenic inputs from the Iberian margin (Gourain et al., 2018; Lemaitre et al., in prep.). The greater proportion of larger size cells, such as diatoms and haptophytes, just like the presence of lithogenic particles, can have increased the sinking speed of the organic matter, leading to a greater POC export flux.

*The west European basin*

Relatively high POC export fluxes were observed at Stations 21 and 26, reaching respectively $4.8 \pm 0.8$ and $7.9 \pm 5.0$ mmol m$^{-2}$ d$^{-1}$. The sampling was performed during the bloom, and the highest PP peak along the section was observed at Station 21 (Fig. 4 and 5) just before the sampling. Station 26 was also sampled after a first bloom peak, and these prior-sampling and high PP peaks could have promoted these high exports. These stations were also characterized by an important proportion of micro-phytoplankton communities which could also explain the high POC exports. Diatoms, known for strongly ballasting the POC exports with their dense frustules (Klaas and Archer, 2002), were dominating. The resulting fast-sinking particles could have promoted the relatively high POC export fluxes (Lemaitre et al., in prep.). For the same area, other studies reported similar POC export fluxes during (May; Thomalla et al., 2008), or just after (July-August; Lampitt et al., 2008; Le Moigne et al., 2013) our sampling period. However, Buesseler et al. (1992) have determined much higher POC fluxes (up to 41 mmol m$^{-2}$ d$^{-1}$) in April-May, during the North Atlantic Bloom Experiment, highlighting an important temporal variability in this basin.

*The Icelandic basin*

One of the highest POC export flux along the transect was determined at Station 32, reaching $8.3 \pm 0.5$ mmol m$^{-2}$ d$^{-1}$ while the POC flux at Station 38 was $4.8 \pm 0.4$ mmol m$^{-2}$ d$^{-1}$. Both stations were sampled during the productive period, although the peak of the bloom was not yet reached (Fig. 4). Nevertheless, the PP at Station 38 remained rather low along the season (Fig. 4) possibly explaining the lower POC export there compared to Station 32. Both stations were dominated by haptophytes, including coccolithophorids (Tonnard et al., in prep.). Their calcium carbonate shells have been shown to promote the export of POC (Francois et al., 2002; Lam et al., 2011; Lemaitre et al., in prep.). In this basin, studies reported higher POC export fluxes, up to 52 mmol m$^{-2}$ d$^{-1}$ (Ceballos-romero et al., 2016; Giering et al., 2016; Martin et al., 2011; Sanders et al., 2010).

*The Irminger basin*



At higher latitudes, diatoms were dominating the phytoplankton communities. The Irminger basin (Stations 44 and 51)
was sampled close to the bloom maximum but, unlike the west European and Icelandic basins, the Irminger basin was
characterized by low POC export fluxes ($1.4 \pm 0.5$ and $2.7 \pm 0.3$ mmol m$^{-2}$ d$^{-1}$, respectively), probably reflecting an
accumulation phase of biomass rather than an export phase. Indeed, this area was also characterized by a high
proportion of particulate $^{234}$Th in surface waters (reaching 94% of the total $^{234}$Th activity at Station 44) and by a very
low P/J ratio indicating that $^{234}$Th was retained in the upper waters rather than exported (Fig. 6; Table 1). In the
literature, a relatively large range of POC export fluxes has been observed in this basin. Puigcorbé et al. (2017)
observed POC export fluxes ranging from 1.5 to 43 mmol m$^{-2}$ d$^{-1}$. Ceballos-Romero et al. (2016) also determined much
higher POC fluxes compared to those observed in the present study, with differences reaching factors of 27 and 19 the
month before and after our sampling, respectively.
*The Labrador basin*
High POC exports were observed within the Labrador basin and in particular at Station 69 where POC export flux
reached $10 \pm 1$ mmol m$^{-2}$ d$^{-1}$. This basin was dominated by micro-phytoplankton species, such as diatoms, and was
sampled just shortly after the peak of PP, indicating the beginning of the decline of the bloom. The combination of the
important PP a few weeks before our sampling and the decline of the diatom bloom likely triggered the high POC
export fluxes, as observed elsewhere (Martin et al., 2011; Roca-Marti et al., 2016b; Stange et al., 2016). As for the
Irminger basin, Puigcorbé et al. (2017) determined a low POC export (0.7 mmol m$^{-2}$ d$^{-1}$) the month before our sampling
period, while Moran et al. (2003) observed higher fluxes reaching 47 mmol m$^{-2}$ d$^{-1}$ in July, one month after our
sampling period.
Overall, POC exports varied strongly along the transect with a factor of 8.6 between the highest and the lowest POC
export flux. The magnitude of these fluxes seems to be dependent on the phytoplankton community structure and thus
on the particle composition and density (Buesseler, 1998; Francois et al., 2002; Honjo, 1996; Lam et al., 2011). The
influence of the ballast effect on POC export is discussed in more details in a companion paper (Lemaitre et al., in
prep.). We have also shown that the magnitude of the POC exports were dependent on the evolution of the bloom, with
high exports during post bloom periods. Studies using deep sediment traps showed these high export events were
driven by large and rapidly sinking aggregates (Lampitt et al., 2001, 2010; Turner, 2002). The comparison with the
literature also suggests that POC export fluxes are strongly variable temporally. Indeed, in only 1 month time lag, POC
fluxes can vary up to a factor of 27 (Ceballos-Romero et al., 2016; Fig. 7) confirming the fast changes of the
biogeochemical parameters controlling the sinking particles and thus the export fluxes.
**4.6. Export and transfer efficiencies of POC**
In order to study the biological carbon pump in the North Atlantic, we used two parameters: the export efficiency
(ThE), which is calculated by dividing the POC export flux at Eq by the PP (Buesseler, 1998) and the transfer efficiency
(T100) which is calculated by the POC export flux at 100 m below Eq divided by the POC export flux at Eq (Fig. 8).
Note that the POC export flux at Eq+100 (Table 3) has been calculated by multiplying the $^{234}$Th flux at Eq+100 by the
POC to $^{234}$Th ratio of large particles at this same depth.





Considering the *in-situ* PP (Table 3), ThE ranged from 1 (Station 44) to 38% (Station 69) with a median value of 7%
along the transect. The highest export efficiencies were determined at Stations 1 and 69 where ThE reached 35 and
38%, respectively. Other stations were characterized by ThE ≤ 14% with a higher range (7 – 14%) at Stations 32, 38,
64 and 77. Export efficiencies around 10% are common in the open ocean (Buesseler, 1998). Lower export efficiency
can be related to important microbial and zooplankton grazing activities as to biomass accumulation in surface waters
(Planchon et al., 2013, 2015). On the contrary, ratios greater than 10% highlight an efficient export of the PP out of
the surface layer. High ThE can be caused by many processes such as the presence of large and/or dense and fast
sinking particles, low surface remineralisation, active zooplankton migration or nutrient stress (Ceballos-romero et al.,
2016; Le Moigne et al., 2016; Planchon et al., 2013). Interestingly, stations with the highest ThE were also
characterized by the lowest PP (Stations 1 and 69) while stations with the lowest ThE were characterized by the highest
PP (Stations 44 and 51). This inverse relationship between PP and ThE was significant for all stations of the GEOVIDE
cruise (regression slope: -0.20; r²=0.58; p<0.01; n=11; Fig. S2) and has been explained in the Southern Ocean by the
temporal decoupling between PP and export (Henson et al., 2015), biomass accumulation in surface waters (Planchon
et al., 2013), and other processes such as zooplankton grazing and bacterial activities (Maiti et al., 2013; Le Moigne et
al., 2016; Roca-Marti et al., 2016a). Indeed, efficient recycling of particles in the upper waters has been observed in
the North Atlantic due to high microbial or grazing activities (Collins et al., 2015; Giering et al., 2014; Marsay et al.,
2015) limiting the POC export to the deep ocean. A recent study in the Icelandic and Irminger basins highlighted the
importance of the bloom dynamics on the particle export efficiency suggesting a strong seasonal variability of the ThE
(Ceballos-Romero et al., 2016). Our estimates are generally in the lower range of export efficiencies reported for the
North Atlantic with values ranging from 1 to 42% in the western European basin (Buesseler et al., 1992; Lampitt et
al., 2008; Thomalla et al., 2008), from 5 to 8% in the Icelandic basin (Ceballos-romero et al., 2016), from 4 to 16% in
the Irminger basin (Ceballos-romero et al., 2016) and from 4 to >100% in the Labrador basin (Moran et al., 2003).
This large range confirms that export efficiencies are highly variable with time and that the North Atlantic during the
period of our study seemed to behave like most of the highly productive areas of the world's ocean, with a low export
efficiency.
However, the ThE calculation is based on two parameters that are integrating processes over different time scales: 24
h for *in-situ* PP and several weeks for export. As a result of this temporal mismatch and due to the strong variability in
PP, ThE ratios were also estimated using the satellite-derived 8-day, 32-day and seasonal PP (Table 3). As seen in
Section 3.6, there are no significant differences between the satellite PP estimates regardless of the integrations times,
and thus no significant differences between the ThE values, except at Stations 1 and 69. Indeed, the ThE values
decrease from 35 to 12% and from 38 to 8% respectively at Stations 1 and 69, suggesting an *in-situ* PP unusually low
during our study, leading to an over-estimated ThE. At the other stations along the transect, no significant ThE changes
were observed regardless of the temporal PP integration.
Carbon transfer efficiencies (T100) ranged from 30 (Station 69) to 78% (Station 32). They were characterized by
greater error bars (see Fig. 8) due to the greater incertitude on the deep POC export flux. The highest T100 were
observed within the Icelandic basin at Stations 32 and 38 with values reaching 78 and 74%, respectively. On the
contrary, the lowest T100 values were determined at Stations 1, 13, 21 and 69 (46, 33, 49 and 30%, respectively) and
highlight firstly, an important regional variability, also reported elsewhere (Lam et al., 2011; Lutz et al., 2002), and



secondly, a greater carbon remineralisation between Eq and Eq+100 m at these latter stations. As already discussed for
the R100 values (see Section 4.4), the low T100 values observed in the eastern part of the transect may be explained
by an important bacterial activity, reinforced in warm waters. This efficient recycling is characteristic of tightly coupled
regeneration-based microbial food webs of oligotrophic regimes (Karl, 1999; Thomalla et al., 2006), such as at Stations
1, 13 and, to a lesser extent, at Station 21. In the Icelandic basin (Stations 32 and 38), the high T100 may be related to
the important abundance of coccolithophorids (Tonnard et al., in prep.) known to enhance the POC export flux to the
deep ocean by ballast effect (Francois et al., 2002; Lam et al., 2011). Indeed, Bach et al. (2016) highlighted that a
bloom of coccolithophorids can increase the transfer efficiency through the mesopelagic layer by 14-24%. Finally, the
Labrador and Irminger basins exhibit relatively similar T100 (between 50 and 69%), except at Station 69 where we
determined the lowest T100. This is also in agreement with the highest R100 (Section 4.4) and carbon remineralisation
flux determined with the $Ba_{xs}$ proxy (Lemaitre et al., 2018).
**5.    Conclusion**
As part of the GEOTRACES program, the GEOVIDE GA01 section allowed us to investigate the [234]Th and POC
export fluxes across the North Atlantic during spring 2014.
The export of [234]Th through sinking particles was particularly efficient at stations close to the margins and in the
Icelandic basin (Stations 1, 64 and 38), while the Irminger basin was characterized by an important retention of [234]Th
in surface waters. This could be due the development of the bloom leading to an accumulation phase of biomass rather
than an export phase. Close to the margins, the abundance of lithogenic particles may have enhanced the [234]Th
scavenging and its subsequent removal to deeper levels in the water column. [234]Th fluxes were also calculated 100 m
below Eq in order to investigate remineralisation processes. [234]Th attenuation appeared more intense at Stations 13 and
21 characterized by warm waters, reinforcing the bacterial activity, and in the Labrador Sea. In the latter area, the
particulate biogenic $Ba_{xs}$ proxy also revealed evidence of enhanced mesopelagic remineralisation, especially at Station
69 (Lemaitre et al., 2018).
The carbon export fluxes varied by a factor ~ 9 along the transect highlighting an important spatial variation. In the
North Atlantic, some studies reported similar POC export estimates but some others determined much higher POC
export fluxes, up to a factor of 27 in only 1 month lag, confirming the high temporal variation of the POC export fluxes
in this ocean, as shown by studies using fixed sediment traps (Antia et al., 2001; Billet et al., 1983; Lampitt et al.,
2010; Peinert et al., 2001).
Different factors were identified for controlling the POC export fluxes regionally and temporally:
i)    The magnitude of the POC export flux is directly related to the intensity and the stage of the bloom.

During the bloom, an accumulation of biomass in surface water may induce a limitation of the POC

export fluxes while exports can increase during the decline of the bloom, likely due to increasing numbers

of rapidly sinking particles.

ii)    The phytoplankton size structure might have influenced the magnitude of the POC export fluxes. Indeed,

the only station characterized by pico-phytoplankton communities was characterized by one of the lowest

POC export flux. However, the areas composed by nano- or micro-phytoplankton were both




characterized by high POC export fluxes, indicating that the size structure was not the main factor

influencing the fluxes during GEOVIDE.

iii)  The phytoplankton community seems to impact the particle composition and density, which play a crucial

role on the particulate sinking velocities and thus on the magnitude of the POC export fluxes. The highest

POC export fluxes were determined at stations where diatoms or coccolithophorids dominated,

suggesting the importance of the ballast effect in the North Atlantic (Lemaitre et al., in prep.).

For most stations, the fraction of primary production that is exported from the surface zone (export efficiency) was
≤14%, which is in agreement with the global ocean export efficiency (~10%; Buesseler, 1998). Export efficiency was
also inversely related to primary production, highlighting that the North Atlantic during our study seems to behave like
most of the highly productive areas of the world's ocean, with a low export efficiency. Finally, the fraction of POC
that is not remineralised in the mesopelagic zone (transfer efficiency) fits within the range of measured transfer
efficiencies reported elsewhere (e.g., Black et al., 2017; Buesseler and Boyd, 2009). The highest transfer efficiencies
were determined at stations where coccolithophorids dominated.
**Acknowledgements**
We would like to thank the captain and the crew of the R/V Pourquoi Pas?, the chief scientists Pascale Lherminier and
Géraldine Sarthou, as well as Fabien Perault and Emmanuel De Saint Léger (CNRS DT-INSU), Pierre Branellec,
Michel Hamon, Catherine Kermabon, Philippe Le Bot, Stéphane Leizour and Olivier Ménage (Laboratoire
d'Océanographie Physique et Spatiale) for their technical expertise during ISP and CTD deployments and Catherine
Schmechtig for the GEOVIDE database management. We also acknowledge Emilie Grossteffan, Manon Le Goff,
Morgane Galinari and Paul Tréguer for the analysis of nutrients. Special thanks to Maxi Castrillejo (UAB, Spain),
Catherine Jeandel (LEGOS, France), Virginie Sanial (WHOI, USA), Raphaëlle Sauzède (LOV, France) and Lorna
Foliot (LSCE, France) for their help at sea and for the pump coordination and collaboration. We would also like to
thank Phoebe Lam for providing two modified McLane ISP. Laurence Monin (MRAC, Belgium), David Verstraeten,
Claire Mourgues and Martine Leermarkers (VUB, Belgium) greatly helped during sample processing and element
analysis by ICP-MS and EA-IRMS. Audrey Plante (ULB, Belgium) and Emilie Le Roy (LEGOS, France) are also
acknowledged for helping to count the residual Thorium-234 activities. Satellite primary production data and
visualizations used in this study were produced with the Ocean Productivity website at Oregon State University.
This work was funded by the Flanders Research Foundation (project G071512N), the Vrije Universiteit Brussel
(Strategic Research Program, project SRP-2), the French ANR Blanc GEOVIDE (ANR-13-BS06-0014), ANR
RPDOC BITMAP (ANR-12-PDOC-0025-01), IFREMER, CNRS-INSU (programme LEFE), INSU OPTIMISP and
Labex-Mer (ANR-10-LABX-19).



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




**Table 1:** Summary of the [234]Th export and scavenging fluxes using steady state (SS) and non-steady state (NSS) models. The
[234]Th export fluxes using the SS model are calculated at the depths corresponding to the bottom of the primary production
zone (PPZ), the equilibrium (Eq) depth and 100 m below Eq (Eq+100); the latter being used to estimate a remineralisation
flux of [234]Th (R100). Negative R100 values indicate an increase of the export flux between Eq and Eq+100. Note that the
depth was fixed to 100 m at Station 26 because of the lower sampling vertical resolution. Consequently, the export flux at
Eq+100 and the R100 were not determined at Station 26.

| Basin | Station | | Export depth m | Th export (SS) dpm m$^{-2}$ d$^{-1}$ | | | Th export (NSS) dpm m$^{-2}$ d$^{-1}$ | | | Th scavenging (SS) dpm m$^{-2}$ d$^{-1}$ | | |
|---|---|---|---|---|---|---|---|---|---|---|---|---|
| Iberian | 1 | PPZ | 155 | 1327 | ± | 137 | | | | | | |
| | | Eq | 90 | 1264 | ± | 104 | 1442 | ± | 80 | 1509 | ± | 189 |
| | | Eq+100 | 190 | 1348 | ± | 199 | | | | | | |
| | | R100 | | -84 | ± | 224 | | | | | | |
| | 13 | PPZ | 82 | 1247 | ± | 99 | | | | | | |
| | | Eq | 110 | 1418 | ± | 111 | 1588 | ± | 86 | 2898 | ± | 285 |
| | | Eq+100 | 210 | 1008 | ± | 187 | | | | | | |
| | | R100 | | 410 | ± | 218 | | | | | | |
| West European | 21 | PPZ | 82 | 1723 | ± | 82 | | | | | | |
| | | Eq | 110 | 1873 | ± | 97 | 2352 | ± | 70 | 3917 | ± | 212 |
| | | Eq+100 | 210 | 1513 | ± | 235 | | | | | | |
| | | R100 | | 360 | ± | 255 | | | | | | |
| | 26 | PPZ | 95 | 1432 | ± | 117 | | | | | | |
| | | Fixed | 100 | 1486 | ± | 117 | 1968 | ± | 98 | 2839 | ± | 220 |
| Icelandic | 32 | PPZ | 75 | 1455 | ± | 92 | | | | | | |
| | | Eq | 130 | 2282 | ± | 1$à19 | 3540 | ± | 113 | 3690 | ± | 199 |
| | | Eq+100 | 230 | 2200 | ± | 227 | | | | | | |
| | | R100 | | 81 | ± | 256 | | | | | | |
| | 38 | PPZ | 70 | 1136 | ± | 80 | | | | | | |
| | | Eq | 80 | 1134 | ± | 95 | 2345 | ± | 115 | 1495 | ± | 160 |
| | | Eq+100 | 180 | 949 | ± | 151 | | | | | | |
| | | R100 | | 185 | ± | 178 | | | | | | |
| Irminger | 44 | PPZ | 37 | 321 | ± | 66 | | | | | | |
| | | Eq | 40 | 321 | ± | 66 | 516 | ± | 90 | 1802 | ± | 71 |
| | | Eq+100 | 140 | 454 | ± | 114 | | | | | | |
| | | R100 | | -132 | ± | 132 | | | | | | |
| | 51 | PPZ | 37 | 495 | ± | 67 | | | | | | |
| | | Eq | 100 | 922 | ± | 103 | 1625 | ± | 108 | 2189 | ± | 260 |
| | | Eq+100 | 200 | 873 | ± | 114 | | | | | | |
| | | R100 | | 49 | ± | 154 | | | | | | |
| Labrador | 64 | PPZ | 83 | 853 | ± | 129 | | | | | | |
| | | Eq | 80 | 855 | ± | 95 | 1423 | ± | 122 | 1142 | ± | 192 |
| | | Eq+100 | 180 | 733 | ± | 200 | | | | | | |
| | | R100 | | 123 | ± | 221 | | | | | | |
| | 69 | PPZ | 35 | 684 | ± | 57 | | | | | | |
| | | Eq | 40 | 758 | ± | 57 | 1068 | ± | 53 | 1257 | ± | 112 |
| | | Eq+100 | 140 | 357 | ± | 148 | | | | | | |
| | | R100 | | 401 | ± | 159 | | | | | | |
| | 77 | PPZ | 55 | 693 | ± | 77 | | | | | | |
| | | Eq | 60 | 696 | ± | 77 | 1169 | ± | 75 | 1529 | ± | 148 |
| | | Eq+100 | 160 | 444 | ± | 146 | | | | | | |
| | | R100 | | 252 | ± | 165 | | | | | | |



**Table 2: Comparison of the steady state POC export fluxes at Eq as determined using the POC:[234]Th ratios in the large**
**(LSF; > 53 μm) and small size fraction (SSF; 1-53 μm).**

| Station # | LSF POC flux mmol m$^{-2}$ d$^{-1}$ | | | SSF POC flux mmol m$^{-2}$ d$^{-1}$ | | |
|---|---|---|---|---|---|---|
| 1 | 12 | ± | 22 | 6.9 | ± | 2 |
| 13 | 2.2 | ± | 0.3 | 3.3 | ± | 0.6 |
| 21 | 4.8 | ± | 0.8 | 6.3 | ± | 1.4 |
| 26 | 7.9 | ± | 5.0 | 6.1 | ± | 3.7 |
| 32 | 8.3 | ± | 0.5 | 8.8 | ± | 0.5 |
| 38 | 4.8 | ± | 0.4 | 5.2 | ± | 0.7 |
| 44 | 1.4 | ± | 0.5 | 2.4 | ± | 0.5 |
| 51 | 2.7 | ± | 0.3 | 3.8 | ± | 0.5 |
| 64 | 7.8 | ± | 1.5 | 5.5 | ± | 4.9 |
| 69 | 10 | ± | 1 | 13 | ± | 1 |
| 77 | 6.1 | ± | 1.5 | 7.5 | ± | 0.9 |







**Table 3:** POC (particulate organic carbon) to $^{234}$Th ratios (in µmol dpm$^{-1}$), POC export fluxes (in mmol m$^{-2}$ d$^{-1}$) at the Eq depth, *in-situ* PP (Fonseca-Batista et al., 2018 and
this study) and satellite-derived PP from the Vertically Generalized Production Model (VGPM) integrated over 8 days, 32 days and over the whole season (in mmol m$^{-2}$ d$^{-}$
$^{1}$) and the POC fluxes at Eq+100 m (in mmol m$^{-2}$ d$^{-1}$). Because of the lower vertical sampling resolution at Station 26, no POC export flux was determined at Eq+100. *The
sampling to determine the *in-situ* PP at Station 51 occurred 24h after the sampling of the particulate $^{234}$Th and POC.

| Station | POC:$^{234}$Th at Eq | POC flux at Eq | *in-situ* PP | 8-days VGPM based PP | 32-days VGPM based PP | seasonal VGPM based PP | POC flux at Eq+100 |
|---|---|---|---|---|---|---|---|
| # | µmol dpm$^{-1}$ | mmol m$^{-2}$ d$^{-1}$ | mmol m$^{-2}$ d$^{-1}$ | mmol m$^{-2}$ d$^{-1}$ | mmol m$^{-2}$ d$^{-1}$ | mmol m$^{-2}$ d$^{-1}$ | mmol m$^{-2}$ d$^{-1}$ |
| 1 | 9 ± 17 | 12 ± 22 | 33 ± 2 | 76 ± 3 | 80 ± 11 | 96 ± 62 | 5.3 ± 23.2 |
| 13 | 1.6 ± 0.2 | 2.2 ± 0.3 | 79 ± 3 | 64 ± 7 | 72 ± 18 | 81 ± 63 | 0.7 ± 0.2 |
| 21 | 2.6 ± 0.4 | 4.8 ± 0.8 | 135 ± 2 | 161 ± 21 | 260 ± 97 | 201 ± 119 | 2.3 ± 0.4 |
| 26 | 5.3 ± 3.3 | 7.9 ± 5.0 | 174 ± 19 | 77 ± 14 | 74 ± 19 | 112 ± 59 | |
| 32 | 3.6 ± 0.1 | 8.3 ± 0.5 | 105 ± 11 | 105 ± 7 | 95 ± 13 | 87 ± 13 | 6.5 ± 0.7 |
| 38 | 4.2 ± 0.1 | 4.8 ± 0.4 | 68 ± 7 | 82 ± 5 | 94 ± 34 | 109 ± 32 | 3.5 ± 0.6 |
| 44 | 4.4 ± 1.3 | 1.4 ± 0.5 | 137 ± 2 | 89 ± 3 | 110 ± 65 | 101 ± 66 | 0.8 ± 0.4 |
| 51 | 2.9 ± 0.01 | 2.7 ± 0.3 | *166 ± 32 | 95 ± 7 | 125 ± 118 | 125 ± 118 | 1.7 ± 0.2 |
| 64 | 9.2 ± 1.1 | 7.8 ± 1.5 | 54 ± 18 | 59 ± 18 | 109 ± 115 | 103 ± 122 | 4.9 ± 1.5 |
| 69 | 14 ± 0.04 | 10 ± 1 | 27 ± 5 | 108 ± 8 | 134 ± 80 | 134 ± 80 | 3.1 ± 1.3 |
| 77 | 8.8 ± 1.9 | 6.1 ± 1.5 | 80 ± 21 | 108 ± 8 | 134 ± 80 | 134 ± 80 | 3.1 ± 1.3 |






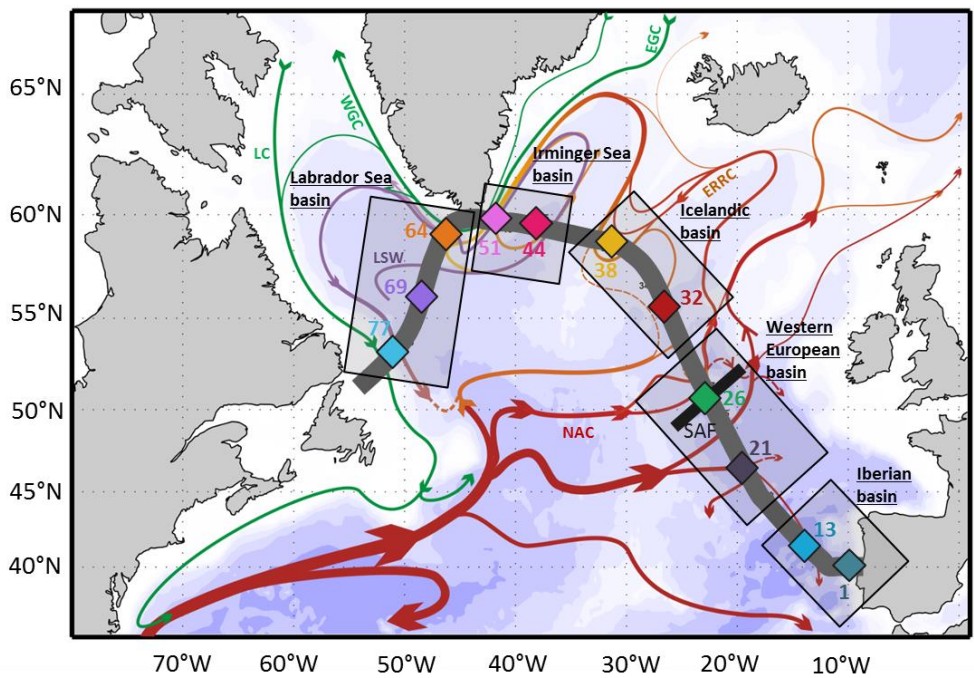


**Figure 1:** Simplified schematic of the surface circulation in the North Atlantic (adapted from Daniault et al., 2016) superimposed with the GEOVIDE cruise track (thick grey line) and stations (colored diamonds). Main surface currents are indicated: East Greenland Current (EGC), West Greenland Current (WGC), Labrador Current (LC), Eastern Reykjanes Ridge Current (ERRC), North Atlantic Current (NAC). The Sub-Arctic Front (SAF) and the Labrador Seawater (LSW) when in surface (i.e. within the Labrador basin) are also represented. Station colors are reused in the following figures.





998

Figure 2: Profiles of the total $^{234}$Th (closed blue circles), total $^{238}$U (thick grey vertical line) and particulate $^{234}$Th activities for the small size fraction (SSF; 1-53 µm; open diamonds) and for the large size fraction (LSF; >53 µm; open triangles). All activities are expressed in dpm L$^{-1}$. The horizontal black line is the Eq depth (depth where $^{234}$Th returns to equilibrium with $^{238}$U), and the horizontal green line is the depth of the PPZ (primary production zone). Error bars are smaller than the size of the symbols





**Figure 3:** Profiles of the POC:$^{234}$Th ratios (µmol dpm$^{-1}$) in the SSF (open symbols) and LSF (closed symbols). The Eq depth, where $^{234}$Th is back to equilibrium with $^{238}$U, is indicated with the grey horizontal line. The thin black line represents the power law fit (POC:$^{234}$Th=a×Z$^{-b}$) of the LSF.

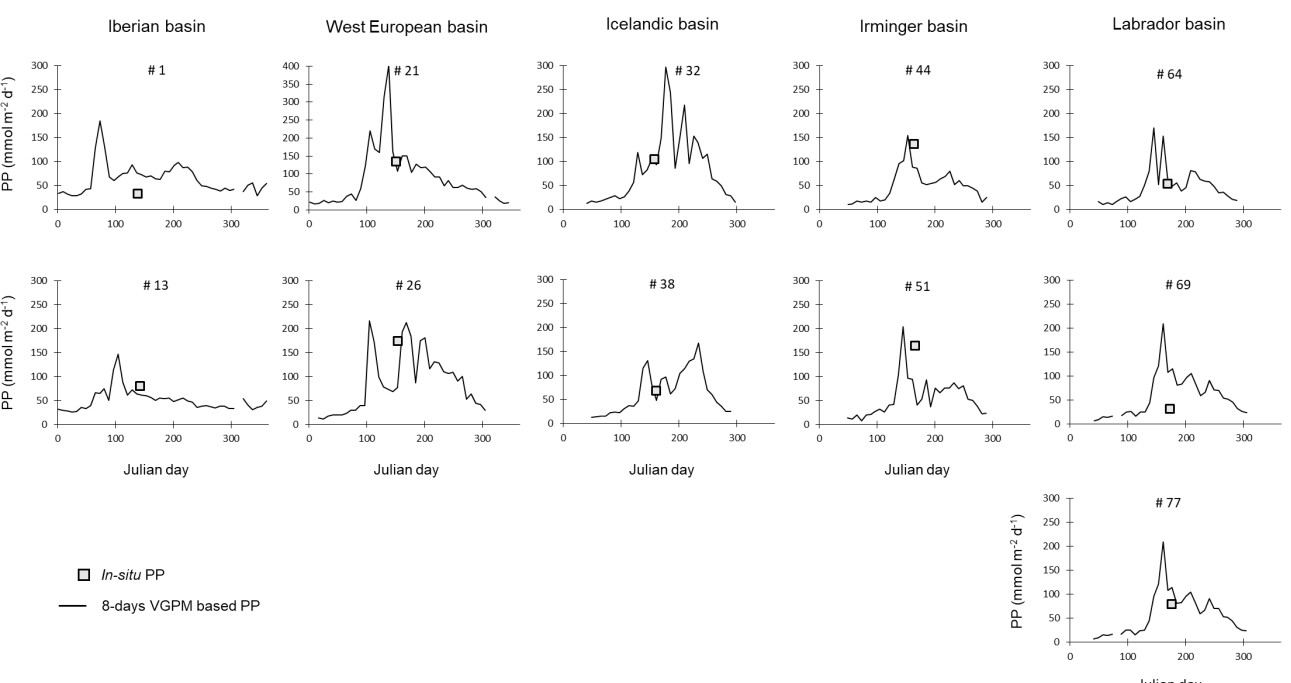


**Figure 4:** *In-situ* (squares) and satellite-derived (continuous lines) primary production (PP; in mmol m⁻² d⁻¹) data at the time of our sampling and along the year 2014.





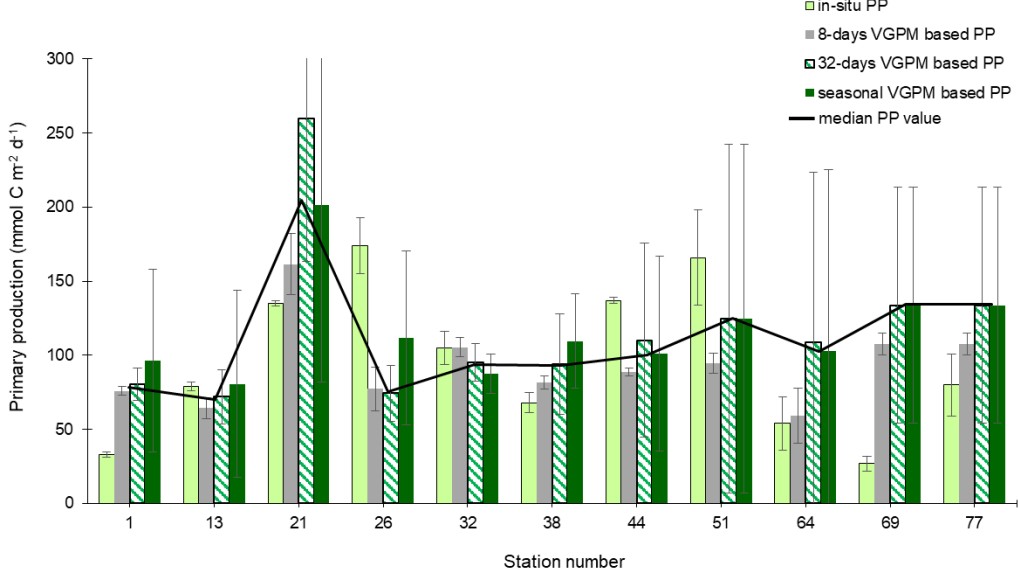

**Figure 5:** Comparison of *in-situ* and satellite (8-days, 32-days and seasonal averages) primary productivities (mmol C m$^{-2}$ d$^{-}$
$^{1}$) along the GEOVIDE transect. The median value is also indicated (black line).





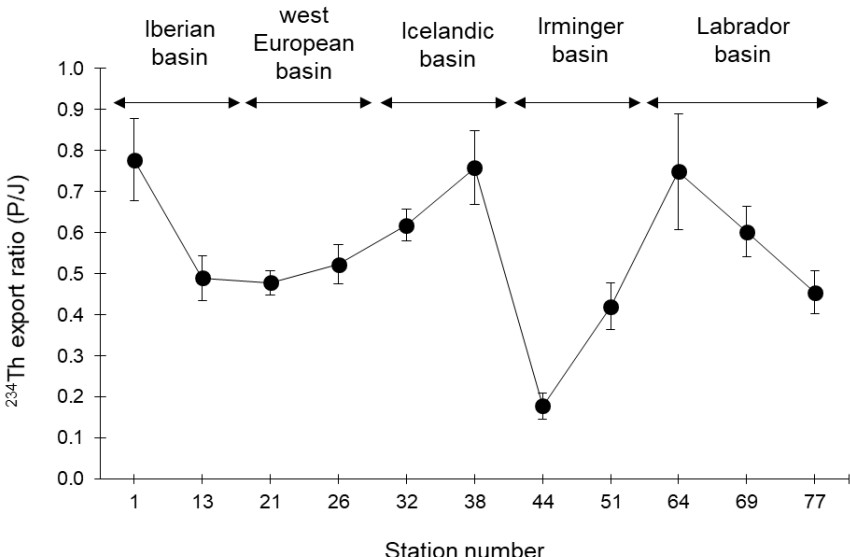


**Figure 6:** Variability of the $^{234}$Th export ratio (i.e., the ratio of the $^{234}$Th export flux over the $^{234}$Th scavenged flux; P/J ratio) along the GEOVIDE section.





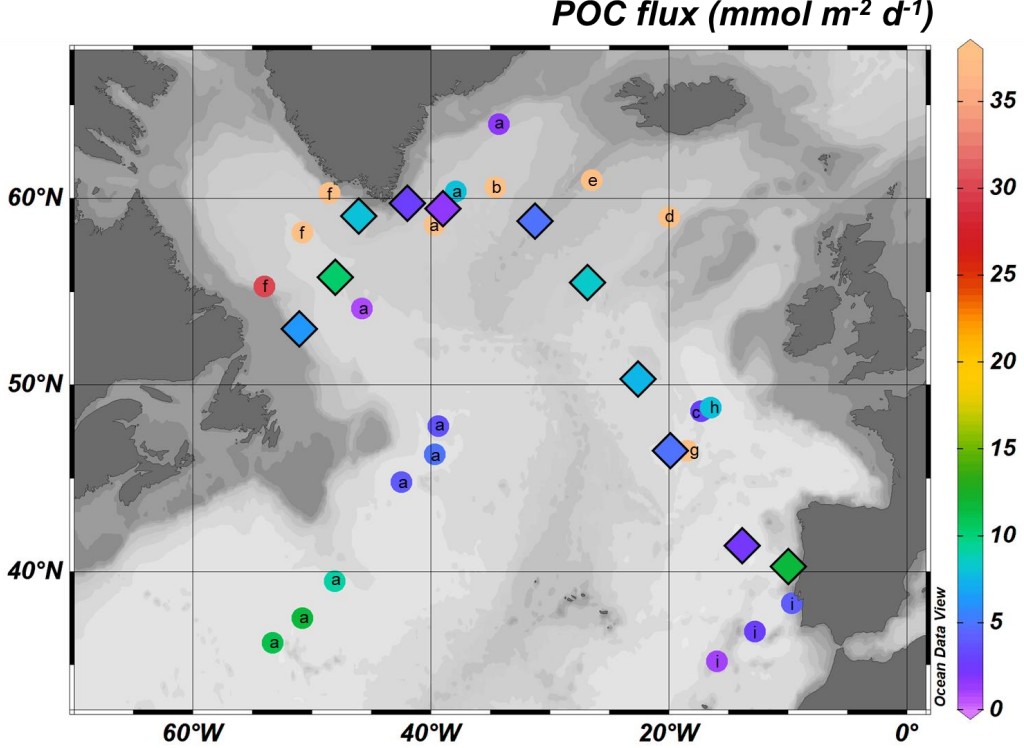


**Figure 7:** Comparison of the POC export fluxes from this study (diamonds with black borders) with other [234]Th-derived
estimates of POC exports in the North Atlantic (a: Puigcorbé et al., 2017; b: Ceballos-Romero et al., 2016; c: Thomalla et
al., 2008; d: Sanders et al., 2010; e: Martin et al., 2011; f: Moran et al., 2003; g: Buesseler et al., 1992; h: Le Moigne et al.,
2013; i: Owens et al., 2015).






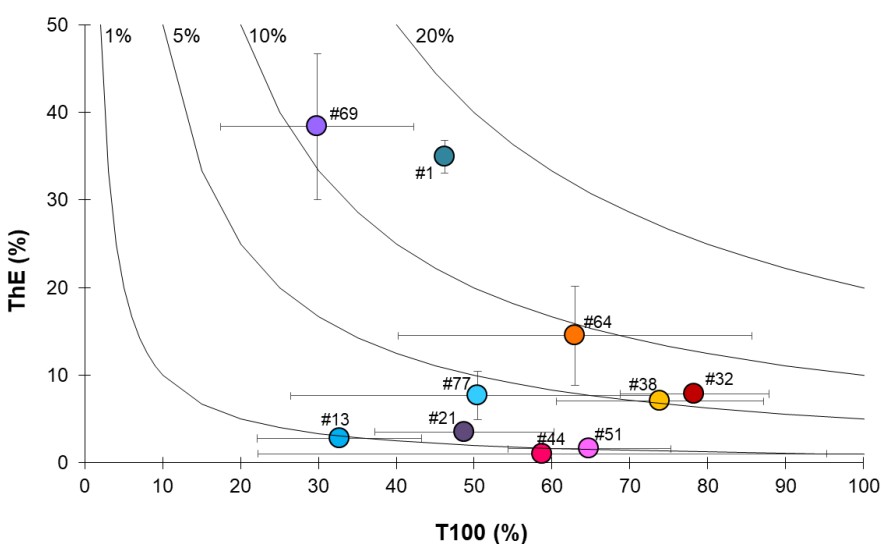


**Figure 8:** Export efficiency (ThE = Export at Eq / *in-situ* PP) versus transfer efficiency (T100 = Export flux at Eq+100 /
**Export flux at Eq). The black lines represent the modelled 1, 5, 10 and 20% of PP exported to depths > Eq+100 m.**