# Peer review of "High variability of particulate organic carbon export along the North Atlantic GEOTRACES section GA01 as deduced from"

_Biogeosciences, 2018_

## Referee Comment (RC1) · Anonymous Referee #1 · 15 Jun 2018

**Review of the manuscript "High variability of export fluxes along the North Atlantic GEOTRACES section GA01: Particulate organic carbon export deduced from the $^{234}$Th method" by Lemaitre et al., submitted to Biogeosciences.**

**Summary**

In this work, Lemaitre et al. present $^{234}$Th-derived POC export fluxes from a Geotraces section in the North Atlantic. They step the reader through the process of obtaining the $^{234}$Th export fluxes that will be converted to POC export fluxes using particulate C/$^{234}$Th ratios. The authors also provide export efficiencies using in situ and satellite-based primary production, and they also examine the transfer efficiencies along the section. The results indicate regional differences based on the dominant phytoplankton groups and the sampling time referred to the peak of the bloom. Despite not having the possibility of resampling the same location, the authors assess the potential differences between steady and non-steady state models and they also make an effort to provide an estimate of the impact that physical fluxes (advection and diffusion) would have had on the $^{234}$Th export fluxes (and by extension, to the derived POC fluxes). I have some concerns with the scavenging flux calculations but they just need to clarify some aspects and discuss potential limitations (see extended comment below). Apart from that, the manuscript is nicely written and it contains a valuable dataset that will contribute to the body of literature using $^{234}$Th to derive POC export fluxes to help characterize the strength and efficiency of the biological carbon pump, particularly in the North Atlantic. With some minor revisions and a bit more of discussion in certain aspects, this manuscript will be a good fit for publication in Biogeosciences.

**Minor comments:**

L46-49: There are two sentences that are repeated

Methods: I understand that Chl-a, phytoplankton community and nutrients (macro and micro) data are obtained from other studies, properly cited within the manuscript. However, I would have liked to see a small paragraph summarizing the methods used to obtain those datasets, particularly considering that there is a full section (2.1) (which is not really methods but more of a description of the study area), where all these nutrient, phytoplankton and chlorophyll-a data is used. Adding a few lines would make the reader's life easier by not needing to look for those papers. Also, a large part of the information included in 2.1 is also mentioned in the discussion, so the authors might want to consider deleting that section, then no needing to include the methods for those analyses.

L87-88: This statement is a bit vague, hard to quantify. Nanophytoplankton species seem to dominate but in the next sentence the emphasis is on picophytoplankton. Also, what do the

authors consider when they say "dominate"? How much higher is the percentage of nanophytoplankton to consider that they are dominant? Above 50%?

L94-95: "Moderate $NO_3$" and then writing $\geq 1$ $\mu$M, which does not have an upper limit, might not be appropriate.

L128: How good was the agreement between the deep $^{234}$Th samples and the $^{238}$U concentrations derived from salinity at those depths?

L131 (and elsewhere in this section): I appreciate the detail in providing the volumes of the spikes and carriers added, however, without the concentrations of those solutions, the information about the volumes added is not really necessary.

L190: "only 10% of the surface value", should be "10% of its maximum value"

2.5 Scavenging fluxes of $^{234}$Th:
I am a bit concerned about the assumptions taken for the scavenging fluxes. In this section, the authors present the equations that have been used to obtain those scavenging fluxes but I think there is information lacking. It is not explained how the dissolved and particulate fractions are obtained: How did the authors obtained the dissolved fraction? Did they subtract the particulate fraction from the total to get the dissolved fraction? Which particulate fraction did they use, the sum of the small and the large particles from the in situ pumps? All this information should be included. Section 4.3 discusses export and scavenging fluxes but my doubts still persist.

I am concerned about the potential limitations because, unless I missed something, the total $^{234}$Th was collected from the CTD rosette, and the particulate $^{234}$Th fraction came from in situ pumps. These are two different sampling methods that could lead to differences when looking at the particulate fraction.

Did the authors calculated the scavenging fluxes using both equations, 8 and 9? In L281 looks like they did but for equation 9 the authors can use the particulate $^{234}$Th, obtained directly from the in situ pumps, but for equation 8, again unless I am missing something, they should subtract that particulate fraction from the total $^{234}$Th to obtain the dissolved fraction of $^{234}$Th.

In summary, I think this section should provide more information to fully understand the calculations done and assess their robustness.

Some small details also from this section:
Eq. 6: The term $V$ has been explained in eq. 2, and even though is quite obvious, maybe point out the fact that the subscript $d$ refers to dissolved (same for the subscript $p$ referring to particulate)

L214: "Equation 5 becomes" it should be "Equation 6 becomes"

L225-226: Could the authors provide the depths for the 0.2% of surface PAR to get an idea about down to what depth is the PP being estimated? Is it more or less close to the depth where the $^{234}$Th fluxes are being calculated?

L245: Could you provide more information regarding "the whole productive period"? How was it defined?

L263: At St 26 the Eq depth in Fig 2 is placed at 100 m but it looks like the deficits goes further down and it reaches equilibrium at about 200 m, but there is lower vertical resolution. Table 1 caption mentions Station 26 has a fixed depth, maybe do the same for the caption of Figure 2.

L264: In this line, the definition of PPZ is correct, mentioning its maximum and not the surface value, as done in line 190, however citing Owens et al., 2015 would probably be more appropriate since the work by Marra et al., 2014 does not use the term Primary Production Zone, as used in this manuscript, although they show that in fact, 1% light level (common definition of the euphotic zone depth) might not be deep enough to reach the compensation depth.

Figure 2: Check Eq depth for St 26 or add explanation in the caption (see comment L263). Both, $^{238}$U and $^{234}$Th symbols (or line, for U) are quite thick and it is hard to see the uncertainties. I am assuming that they are there, just within the width of the symbol, right? Linked to that aspect, $^{238}$U activities range from 2.19 to 2.53 dpm L$^{-1}$, but it is really hard to tell from Figure 2. Minor thing, the $^{238}$U line for St 77 seems to be clearer than the rest. It could be useful to color code the labels of the stations to match the colors in Fig 1, or to group them by basins, or indicate to which basin they belong to.

L265: Maybe add "e.g." when citing those two studies where they integrate the Th deficits to the PPZ since there are a few more published studies that have used that same approach.

Figure 3: The uncertainties of the POC to $^{234}$Th ratios are not shown on the graph but there are uncertainties reported for POC and $^{234}$Th separately in Table S2. It looks like the uncertainties have not been considered in the fitting curve. What would the uncertainties of the ratios at Eq. depth be if those uncertainties on the POC and $^{234}$Th content were taken into account when doing the fitting?

L349: The compilations by Le Moigne et al 2013 (global) or Puigcorbe et al 2017 (North Atlantic) include most of the papers cited and will make the citation shorter.

L358: Maybe delete "and argued". Argued is used when one wants to make a point but my guessing is that the authors mean that there is another paper that provides more information.

Also in this line, "details" should be singular (same in L571).

L360: Delete "the" (…PP varied by a factor of…)

L360-369: In some cases PP are presented with uncertainties and sometimes without.

L380-381: Briefly define "productive period" (Is it starting with the PP increase of 30% above winter value mentioned in L449?)

L405-411: The Irminger Basin in spring is a really patchy and dynamic area, as shown by Le Moigne et al (2012) and Puigcorbe et al (2017). The exercise of trying to quantify the impact of physical processes is interesting, however it is a bit of a stretch with just two stations that are also relatively distant. The reference to the Artic and Greenland shelf waters helps to support the author's argument but I think the patchiness (bloom patchiness) during the productive season should also be mentioned (somehow done later on when discussing the bloom stage during the sampling period).

L417: I do not understand the need of the sentence "The vertical advection can also impact the distribution of $^{234}$Th" when previously (L414) there is a sentence that reads as: "the vertical transport of $^{234}$Th associated with small-scale structures could represent up to 20%", it seems redundant.

L487: Maybe reduce the number of references

L495: Similar remineralization although one study was conducted in the tropical Pacific and the other in the North Atlantic Ocean. If the authors want to provide that comparison it might be interesting to discuss a bit the similarities and differences between the studies that lead to comparable values (although some higher values were reported in the tropical Pacific) since one could expect different planktonic communities in both regions, leading to different remineralization intensities.

L509: Stipulate in the following "section"

L583: Maybe specify that the extrapolation curves from Fig 3 were used to obtain the deep POC to $^{234}$Th ratios.

L642-644 (and previously mentioned too): Could the authors provide a potential cause of that enhanced remineralization in the cold waters of the Labrador Sea, especially since the biogenic Ba$_{xs}$ also shows signs of remineralization. Is it also due to bacterial activity? For how it is written it looks like the authors believe is not due to bacterial activity.

L651: This statement is not strictly quantitatively proven and although the authors provide the date of the peak of the bloom and PP values, they do not refer to the intensity of the bloom

(intensity meaning magnitude of PP? Duration of the bloom? Duration of the bloom with sustained high PP values?). The authors discuss the temporality of the bloom with respect to the sampling time, which has been done in previous studies, but it could be interesting to produce a figure or correlation between the stage of the bloom (and/or intensity of the bloom, if defined) and the magnitude of the POC to support this statement in a more quantitative manner to be able to say that they are, in fact, directly related.

L660: I would delete the first sentence of the point iii) of the conclusions because that is not something that has been studied in this manuscript, it is probably going to be done in the coming Lemaitre et al. in prep. manuscript.

---

## Referee Comment (RC2) · Anonymous Referee #2 · 16 Jul 2018

This manuscript presents a valuable dataset that deserves publication. However the manuscript is missing clear motivations and objectives (see comment 1). This short-coming has an impact throughout the manuscript, which is tedious to read and not as informative as it could be (see comment #2). The manuscript is long but new results advertised in the abstract are not clearly highlighted and discussed in the main text (e.g. control from phytoplankton size structure and the stage of the bloom). The result section is a very descriptive listing of all measured parameters and the discussion ressembles a result section (comments #3 and #4). I recommend major revisions to

improve the readability and strengthen the main points.

Major Comments

1) The manuscript lacks a clear objective. In Line 61, "According to the impact of these biogeochemical factors [...], the efficiency of the NAtl to transfer POC .. can be questioned. In this context, we investigated the ... export using Thorium." How is the state-of-the-art presented between L32 to 60 questioning the transfert efficiency established in previous studies? After presenting the state-of-the-art, I strongly encourage the authors to present what open question or inconsistency they are trying to address with their dataset. Possible avenues are: What is missing in previous studies? How is this dataset complementary or inconsistent with previous data? I suggest following the traditional structure: 1_ Previous studies showed that X .... 2_ However, Y is still unknown (or this is inconsistent with Z); 3_ Here, we examine/show/leverage.... This objective should also guide the reader in the result and discussion section (see comment 2).

2) the result section appears as a long list of parameters (e.g. 3.3- Particulate Th and POC distribution, 3.4- POC:Th ratios, 3.5-POC export; 3.6- PP), and include too many methodological details. For example: L 287 to 292 "LSF particles are collected on silver GF/F filters ..."; L329-332; L377 "Using the 8-day average data, PP was estimated for the preceding month and the whole productive period... ". The result section should be re-worked to emphasize the important connections between the different measured parameters (export, PP, planktonic composition etc.). One option would be to present the results per biogeochemical province and make these links. Another option would be to organise the result section based on processes and/or novel findings (for example: Control by phytoplankton size structure, Modulation by stage of the bloom ... or Flux attenuation in mesopelagic zone, which is now in the discussion but would fit better in as a result section- see comment #4). Please move methodological details to method section or remove when it is duplicated.

3) the discussion of uncertainty opens the discussion (sections 4.1 and 4.2 NSS and physical transport). While I value this discussion, it is not novel and has been discussed in previous studies. I suggest to move it to the end of the manuscript. Please start with what is new and motivating before discussing the limitations of the method.

4) large part of the discussion pertains to the result section (sections 4.3, 4.4 and 4.5) and could help organise the results (see comment #2). There are also many method­ological details in the discussion section that should be (re)moved (e.g. L446, L462, L483), in particular when these details are stated several times. For example, L483 explains what thorium deficit is, even though this explanation is already included in the introduction, the method and the result sections. The discussion should emphasize what this study brings to existing studies and discuss the limitations.

5) please streamline the text. Many methodological concepts are presented and intro­duced in several sections (e.d. thorium deficit, PP measurements etc.).
* * *
Minor comments: L43-46: two sentences are repeated in the introduction. Remove one version.

L61-63 needs to be rephrased (see comments above)

Method section could be sharpened by limiting the use of "moderate" (e.g. L91, L106, L116). Please limit the use of "moderate" (e.g. L91, L106, L116), which is rather vague. The word "briefly" (L127, L137, L222) should also be avoided. Either you have described the method and you can remove "briefly", or you haven't described enough and should include additional references or details.

L102: associated with (not to)

L235: other estimates of PP mentioned here could be added for comparison in Figure 4. This would give confidence in the author's choice and inform the reader on the uncertainty associated with these estimates.

L259-261: move definition of PPZ above, when it is first mentioned.

L110: Tonnard et al, in prep and L521 Lemaitre et al, in prep. Does the journal authorized unpublished papers?
* * *

---

## Author Response (AR1)

Dear Editor and Referees,

We first would like to thank the Editor and both referees who provided useful comments to improve our paper entitled "High variability of particulate organic carbon export along the North Atlantic GEOTRACES section GA01 as deduced from $^{234}$Th fluxes".

Please see below our detailed answers to the Referees and to the Editor, including corresponding lines of the revised manuscript. We copied the comments in this document in italics, followed by our answers in black font. The new text that we propose to add in the revised manuscript is in red font.

We hope that you will find the manuscript suitable for publication,

Kind regards,

Nolwenn Lemaitre and co-authors.

**Anonymous Referee 1**

*General comments: Apart from that, the manuscript is nicely written and it contains a valuable dataset that will contribute to the body of literature using $^{234}$Th to derive POC export fluxes to help characterize the strength and efficiency of the biological carbon pump, particularly in the North Atlantic. With some minor revisions and a bit more of discussion in certain aspects, this manuscript will be a good fit for publication in Biogeosciences.*

We would like to thank the referee for these very positive comments. All suggestions have been taken into account and are detailed below.

- *Lines 46-49: There are two sentences that are repeated.*

The introduction section has changed quite a lot, and the repeated sentence has been deleted.

- *Method: I understand that Chl-a, phytoplankton community and nutrients (macro and micro) data are obtained from other studies, properly cited within the manuscript. However, I would have liked to see a small paragraph summarizing the methods used to obtain those datasets, particularly considering that there is a full section (2.1) (which is not really methods but more of a description of the study area), where all these nutrient, phytoplankton and chlorophyll-a data is used. Adding a few lines would make the reader's life easier by not needing to look for those papers. Also, a large part of the information included in 2.1 is also mentioned in the discussion, so the authors might want to consider deleting that section, then no needing to include the methods for those analyses.*

As suggested, section 2.1 has been deleted and has been integrated to the Introduction section where we describe now the different biogeochemical basins. Many details of the section 2.1 have been deleted and for that reason, we decided not to describe the methods to acquire the nutrient concentrations or the pigment data. We are citing publications describing the methods though (nutrient and pigment analyses according to Aminot and Kerouel (2007) and Ras et al. (2008), respectively).

Lines 50-70: The low nutrient availabilities (surface nitrate and silicate concentrations < 1 µmol L$^{-1}$; nutrient analyses according to Aminot and Kérouel, 2007) in the Iberian basin limits the biomass

development giving the opportunity to pico-phytoplankton, such as cyanobacteria, to grow (~ 35% of the total Chl-*a* at Station 13; Tonnard et al., in prep.; pigment analyses according to Ras et al., 2008), a situation which is typical for the North Atlantic subtropical gyre (Moore et al., 2008; Zehr and Ward, 2002). The Iberian basin can also be influenced by a local upwelling, close to the Iberian margin (Costa Goela et al., 2016; Zúñiga et al., 2016; http://marine.copernicus.eu/) and potentially fueling the area with nutrient-rich, but upwelling was not active during GEOVIDE (Shelley et al., 2016).

In the subpolar region, in the Irminger and Labrador basins, phytoplankton growth is strongly light-limited seasonally (Riley, 1957) and the key parameter for alleviating these limitations is the progressive shoaling of the mixed layer. There, micro-phytoplankton, such as diatoms, dominate the phytoplankton bloom (≥ 50% of the total Chl-*a*; Tonnard et al., in prep.). Both basins were influenced by strong hydrodynamic features, such as the Irminger gyre, the Eastern Greenland Current (EGC), the Western Greenland Current (WGC), the Labrador Current (LC; Zunino et al., 2017) and the subduction of the Labrador Seawater (LSW) which was particularly intense (1700 m-deep convection) during the winter 2013-2014 (Kieke and Yashayaev, 2015).

Between the subtropical and subpolar regions, the west European and Icelandic basins represent a transition zone where nutrients and/or light can limit primary production (Henson et al., 2009). During GEOVIDE the silicic acid stock was low (≤ 1 µmol L$^{-1}$) leading to the growth of nano-phytoplankton, such as haptophytes including coccolithophorids (between 45 and 80% of the total Chl-*a*; Tonnard et al., in prep.). This region is influenced by the Eastern Reykjanes Ridge Current (ERRC) and by the North Atlantic Current (NAC) with the southernmost sub-branch evolving in a cyclonic eddy and the sub-arctic front (SAF). SAF separates cold and fresh waters from the subpolar region and the warm and salty waters from the subtropical region (Zunino et al., 2017).

- *Line 87-88: This statement is a bit vague, hard to quantify. Nanophytoplankton species seem to dominate but in the next sentence the emphasis is on picophytoplankton. Also, what do the authors consider when they say "dominate"? How much higher is the percentage of nanophytoplankton to consider that they are dominant? Above 50%?*

The general phytoplankton communities observed along the transect are presented in the Introduction Section. The percentage of the phytoplankton communities relative to the total chlorophyll-a concentration are now stated in the Introduction section. By "dominant community", we mean the most represented community (characterized by the highest percentage relative to the total Chl-a).

Lines 50-54: The low nutrient availabilities (surface nitrate and silicate concentrations < 1 µmol L$^{-1}$; nutrient analyses according to Aminot and Kérouel, 2007) in the Iberian basin limits the biomass development giving the opportunity to pico-phytoplankton, such as cyanobacteria, to grow (~ 35% of the total Chl-*a* at Station 13; Tonnard et al., in prep.; pigment analyses according to Ras et al., 2008), a situation which is typical for the North Atlantic subtropical gyre (Moore et al., 2008; Zehr and Ward, 2002).

Lines 59-60: There, micro-phytoplankton, such as diatoms, dominate the phytoplankton bloom (≥ 50% of the total Chl-*a*; Tonnard et al., in prep.).

Lines 65-67: During GEOVIDE the silicic acid stock was low (≤ 1 µmol L$^{-1}$) leading to the growth of nano-phytoplankton, such as haptophytes including coccolithophorids (between 45 and 80% of the total Chl-*a*; Tonnard et al., in prep.).

The new section 4.3 makes the connection between POC export fluxes and phytoplankton size and community structure, and presents thus more details on the phytoplankton communities observed at each station. For instance, the differences observed within the Iberian basin, between Stations 1 and 13, are clearly stated.

Lines 509-510: Within the Iberian basin, the highest abundance of pico-phytoplankton was observed at Station 13 (Tonnard et al., in prep.). These conditions are typical of the subtropical and oligotrophic waters (Dortch and Packard, 1989).

Lines 514-517: However, Station 1 was characterized by a greater POC export that could be related to the mixed proportion of micro-, nano- and pico-phytoplankton and thus to the greater proportion of larger cells such as diatoms or haptophytes, increasing the particle sinking velocity.

- *Lines 94-95: "Moderate NO$_3^-$" and then writing ≥1 µM, which does not have an upper limit, might not be appropriate.*

Most of the details on nutrient concentrations have been deleted; but when specified in the Introduction section, the upper limits are stated.

Lines 50-51: The low nutrient availabilities (surface nitrate and silicate concentrations < 1 µmol L$^{-1}$; nutrient analyses according to Aminot and Kérouel, 2007) in the Iberian basin…

Lines 65-66: During GEOVIDE the silicic acid stock was low (≤ 1 µmol L$^{-1}$) leading to the growth of nano-phytoplankton …

- *Lines 128: How good was the agreement between the deep $^{234}$Th samples and the $^{238}$U concentrations derived from salinity at those depths?*

The $^{234}$Th/$^{238}$U ratio averaged 1.00 in deep samples, and on 15 deep samples, the standard deviation was about 0.02, highlighting the perfect equilibrium between both radionuclides. Details have been added in Section 2.1.

Lines 106-109: Deep samples (between 1000 and 3500 m) were taken for the calibration of the low level beta counting (Rutgers van der Loeff et al., 2006) based on the knowledge that $^{234}$Th and $^{238}$U are generally in secular equilibrium at such depths (in this study, the deep ocean average $^{234}$Th/$^{238}$U ratio = 1.00 ± 0.02; n=15).

- *Lines 131 (and elsewhere in this section): I appreciate the detail in providing the volumes of the spikes and carriers added, however, without the concentrations of those solutions, the information about the volumes added is not really necessary.*

This is very true. As suggested by reviewer 2, many details, such as the volumes, have been deleted in order to get this manuscript less tedious to read.

- *Line 190: "only 10% of the surface value", should be "10% of its maximum value".*

OK. This has been modified.

 … as well as for z representing the base of the primary production zone (PPZ), i.e. the depth where in-situ fluorescence was only 10% of its maximum value (Owens et al., 2014).

- *2.5 Scavenging fluxes of [234]Th: I am a bit concerned about the assumptions taken for the scavenging fluxes. In this section, the authors present the equations that have been used to obtain those scavenging fluxes but I think there is information lacking. It is not explained how the dissolved and particulate fractions are obtained: How did the authors obtained the dissolved fraction? Did they subtract the particulate fraction from the total to get the dissolved fraction? Which particulate fraction did they use, the sum of the small and the large particles from the in situ pumps? All this information should be included. Section 4.3 discusses export and scavenging fluxes but my doubts still persist.*
  *I am concerned about the potential limitations because, unless I missed something, the total [234]Th was collected from the CTD rosette, and the particulate [234]Th fraction came from in situ pumps. These are two different sampling methods that could lead to differences when looking at the particulate fraction.*

Right, thank you for notifying this. We included the missing information and hopefully the section is clearer now.

To get the dissolved fraction, we subtracted the particulate from the total fraction. The particulate fraction was the sum of the small and large size particulate fractions (SSF+LSF).

The total and particulate fractions were not obtained by the same sampling method. It would be indeed ideal to have both fractions from the same device. However, particulate [234]Th cannot be obtained accurately on low volume samples (i.e., 4L) because of the adsorption of dissolved [234]Th on filter for example (Buesseler et al., 2006).

 To estimate the rate of removal of [234]Th from the dissolved to the particulate form, i.e., the scavenging flux of [234]Th (Coale and Bruland, 1985), we deduced the dissolved [234]Th activities by subtracting the particulate (SSF+LSF) from the total [234]Th activities, keeping in mind, though, that the sampling method for the total and particulate phases differed. Because the sampling resolution was different, total [234]Th data were averaged at the sampling depth of particulate [234]Th.

> *Did the authors calculated the scavenging fluxes using both equations, 8 and 9? In L281 looks like they did but for equation 9 the authors can use the particulate [234]Th, obtained directly from the in situ pumps, but for equation 8, again unless I am missing something, they should subtract that particulate fraction from the total [234]Th to obtain the dissolved fraction of [234]Th.*

The four equations, presented in the previous manuscript were describing the dissolved and particulate activities with a 2-box model. To calculate the scavenging fluxes, we actually use only 2 equations (Eq. 6 and 7, below). In order to get this section clearer and again in order to get a manuscript less tedious to read, we kept the essential equations.

 The mass balance equation for dissolved [234]Th can be written as follows:

$$\frac{dA_{Thd}}{dt} = \lambda A_U - \lambda A_{Thd} - J + V \qquad (6)$$

where $A_{Thd}$ is the activity of dissolved [234]Th in dpm L$^{-1}$; $A_U$ and $\lambda$ are defined in Eq. 2; J is the net removal flux from the dissolved to the particulate form (scavenging flux) in dpm L$^{-1}$ d$^{-1}$; and V is the sum of the advective and diffusive fluxes in dpm L$^{-1}$ d$^{-1}$.

Using again the steady state assumption (dissolved [234]Th activities remain constant over time) and ignoring the physical terms (V), Eq. 6 becomes:

$$J = \lambda \int_0^z (A_U - A_{Thd})dz \qquad\qquad (7)$$

where J in dpm m$^{-2}$ d$^{-1}$ is the net flux of scavenging integrated to the depth z. In our case, the calculation was performed at the Eq depth for comparison with the [234]Th export flux (P in Eq. 3).

*In summary, I think this section should provide more information to fully understand the calculations done and assess their robustness.*

*Some small details also from this section:*
*Eq. 6: The term V has been explained in eq. 2, and even though is quite obvious, maybe point out the fact that the subscript d refers to dissolved (same for the subscript p referring to particulate)*

As both particulate and dissolved fractions are now described in two different sections (2.3 and 2.4), the subscripts have been deleted.

- *Line 214: "Equation 5 becomes" it should be "Equation 6 becomes"*

OK. This has been modified.

- *Line 225-226: Could the authors provide the depths for the 0.2% of surface PAR to get an idea about down to what depth is the PP being estimated? Is it more or less close to the depth where the [234]Th fluxes are being calculated?*

OK. This has been added in section 2.6.

Lines 237-239: Daily PP was then estimated by integrating the uptake rates from the surface down to 0.2% of surface PAR, which was located between 48 and 116 m depending on the station. The 0.2% of surface PAR depth was roughly corresponding to the Eq depth although, at few stations, a 42 m difference was observed.

- *Line 245: Could you provide more information regarding "the whole productive period"? How was it defined?*

OK. This has been added in section 2.7.

Lines 252-253: The whole productive period is the period between the bloom start (defined by a PP increase of 30% above the winter value) and the sampling date (Fig. 5).

- *Line 263: At St 26 the Eq depth in Fig 2 is placed at 100 m but it looks like the deficits goes further down and it reaches equilibrium at about 200 m, but there is lower vertical resolution. Table 1 caption mentions Station 26 has a fixed depth, maybe do the same for the caption of Figure 2.*

OK. The caption of Figure 2 has been modified.

 Profiles of the total $^{234}$Th (closed circles), total $^{238}$U (black dotted vertical line) and particulate $^{234}$Th activities for the small size fraction (SSF; 1-53 µm; open diamonds) and for the large size fraction (LSF; >53 µm; closed triangles). All activities are expressed in dpm L$^{-1}$. The horizontal black line is the Eq depth (depth where $^{234}$Th returns to equilibrium with $^{238}$U), and the horizontal green line is the depth of the PPZ (primary production zone). Error bars are plotted but may be smaller than the size of the symbols. Note that the Eq depth at Station 26 is fixed at 100 m because of the lower sampling vertical resolution.

- *Line 264: In this line, the definition of PPZ is correct, mentioning its maximum and not the surface value, as done in line 190, however citing Owens et al., 2015 would probably be more appropriate since the work by Marra et al., 2014 does not use the term Primary Production Zone, as used in this manuscript, although they show that in fact, 1% light level (common definition of the euphotic zone depth) might not be deep enough to reach the compensation depth.*

Right, the citation has been changed. Moreover, the definition of the PPZ depth is now given only in section 2.3.

Lines 169-176: Eq. 3 has been solved for z taken as the depth (Eq) at the base of the $^{234}$Th deficit zone (Eq = depth where $^{234}$Th activity is back to secular equilibrium with $^{238}$U) as well as for z representing the base of the primary production zone (PPZ), i.e. the depth where in-situ fluorescence was only 10% of its maximum value (Owens et al., 2014). The Eq depth matched relatively well with the PPZ depth, and on average, difference between both was only 16 m, with the largest difference (~ 60 m) at Stations 1, 32 and 51 (Fig. 2). Considering that there can be export (or remineralisation) below or above the PPZ depth, only the export fluxes at the Eq depth will be discussed as they represent the fully-integrated depletion of $^{234}$Th in the upper waters and thus the maximal export.

- *Figure 2: Check Eq depth for St 26 or add explanation in the caption (see comment L263). Both, $^{238}$U and $^{234}$Th symbols (or line, for U) are quite thick and it is hard to see the uncertainties. I am assuming that they are there, just within the width of the symbol, right? Linked to that aspect, $^{238}$U activities range from 2.19 to 2.53 dpm L$^{-1}$, but it is really hard to tell from Figure 2. Minor thing, the $^{238}$U line for St 77 seems to be clearer than the rest. It could be useful to color code the labels of the stations to match the colors in Fig 1, or to group them by basins, or indicate to which basin they belong to.*

Following your suggestions, we modified the figure and its caption. Indeed, all the errors are indicated but may be hidden by the symbols.

[Figure]

Figure 1: Profiles of the total $^{234}$Th (closed circles), total $^{238}$U (black dotted vertical line) and particulate $^{234}$Th activities for the small size fraction (SSF; 1-53 µm; open diamonds) and for the large size fraction (LSF; >53 µm; closed triangles). All activities are expressed in dpm L$^{-1}$. The horizontal black line is the Eq depth (depth where $^{234}$Th returns to equilibrium with $^{238}$U), and the horizontal green line is the depth of the PPZ (primary production zone). Error bars are plotted but may be smaller than the size of the symbols. Note that the Eq depth at Station 26 is fixed at 100 m because of the lower sampling vertical resolution.

- *Line 265: Maybe add "e.g." when citing those two studies where they integrate the Th deficits to the PPZ since there are a few more published studies that have used that same approach.*

This sentence has been removed from the Result section.

- *Figure 3: The uncertainties of the POC to $^{234}$Th ratios are not shown on the graph but there are uncertainties reported for POC and $^{234}$Th separately in Table S2. It looks like the uncertainties have not been considered in the fitting curve. What would the uncertainties of the ratios at Eq. depth be if those uncertainties on the POC and $^{234}$Th content were taken into account when doing the fitting?*

The figure and its caption have been modified. Errors of POC to $^{234}$Th ratios are shown and even if the size of the symbols has been reduced, they may not be visible. In the caption, we now give the median percentage of these errors relative to the value of the ratio.

[Figure]

Figure 2: Profiles of the POC:$^{234}$Th ratios (µmol dpm$^{-1}$) in the SSF (open symbols) and LSF (closed symbols). The Eq depth, where $^{234}$Th is back to equilibrium with $^{238}$U, is indicated with the grey horizontal line. The thin black line represents the power law fit (POC:$^{234}$Th=a×Z$^{-b}$) of the LSF. The median percentage errors on POC:$^{234}$Th ratios are respectively representing 5 and 6% of the value for the SSF and the LSF. Error bars are plotted but may be smaller than the size of the symbols.

The uncertainty of the extrapolated ratio at the Eq depth is deduced from the fit and not from the analytical uncertainties of both POC concentrations and particulate [234]Th activities. The error deduced from the fit is much larger than the one from the analytical error. This has been specified within the manuscript, section 2.5.

Lines 216-221: We estimated POC export fluxes by multiplying the [234]Th export flux with the POC:[234]Th ratio, both determined at the Eq depth. A power law fit was used to determine the POC:[234]Th ratios at Eq (Fig. 3). Errors of the POC:[234]Th ratios extrapolated at the Eq depth are deduced from the power law fit, using a root sum of square method. This error is much larger than analytical errors of both POC concentrations and particulate [234]Th activities. POC fluxes were determined by using the POC:[234]Th ratios of the LSF (> 53 µm) as well as the SSF (1-53 µm) samples, and both estimations were compared (Table 2).

- *Line 349: The compilations by Le Moigne et al 2013 (global) or Puigcorbe et al 2017 (North Atlantic) include most of the papers cited and will make the citation shorter.*

Right, this has been modified.

Lines 225-228: As large and rapidly sinking particles usually drive most of the export (Lampitt et al., 2001; Villa-Alfageme et al., 2016), most of the studies dedicated to POC export fluxes in the North Atlantic used the POC:[234]Th ratios from the LSF (see Le Moigne et al., 2013b; Puigcorbé et al., 2017).

- *Line 358: Maybe delete "and argued". Argued is used when one wants to make a point but my guessing is that the authors mean that there is another paper that provides more information. Also in this line, "details" should be singular (same in L571).*

OK.

- *Line 360: Delete "the" (…PP varied by a factor of…).*

OK.

- *Line 360-369: In some cases PP are presented with uncertainties and sometimes without).*

OK, we removed the uncertainties in the text. They are written in Table 3.

- *Line 380-381: Briefly define "productive period" (Is it starting with the PP increase of 30% above winter value mentioned in L449?).*

OK. This sentence has been removed from the Discussion section and the productive period is defined in section 2.7.

Lines 252-253: The whole productive period is the period between the bloom start (defined by a PP increase of 30% above the winter value) and the sampling date (Fig. 5).

- *Line 405-411: The Irminger Basin in spring is a really patchy and dynamic area, as shown by Le Moigne et al (2012) and Puigcorbe et al (2017). The exercise of trying to quantify the impact*

*of physical processes is interesting, however it is a bit of a stretch with just two stations that are also relatively distant. The reference to the Artic and Greenland shelf waters helps to support the author's argument but I think the patchiness (bloom patchiness) during the productive season should also be mentioned (somehow done later on when discussing the bloom stage during the sampling period).*

This is right, thank you for your comment. The patchiness of the bloom within the Irminger basin is now mentioned when talking about impact of the physical processes.

Lines 415-419: The Irminger basin in spring is a really patchy and dynamic area (Ceballos-romero et al., 2016; Le Moigne et al., 2012; Puigcorbé et al., 2017) but the relatively high variability of the $^{234}$Th fluxes found at these two stations (321 and 922 dpm m$^{-2}$ d$^{-1}$, respectively) may also indicate a potential influence of lateral advection. The higher export flux at Station 51 could reflect an input of $^{234}$Th depleted waters originating from the Arctic and/or the Greenland shelf.

- *Line 417: I do not understand the need of the sentence "The vertical advection can also impact the distribution of $^{234}$Th" when previously (L414) there is a sentence that reads as: "the vertical transport of $^{234}$Th associated with small-scale structures could represent up to 20%", it seems redundant.*

Right, the sentence has been deleted.

- *Line 487: Maybe reduce the number of references.*

Right, this sentence is now in the Introduction section.

Lines 92-94: In the subsurface waters any excess of $^{234}$Th relative to $^{238}$U, is taken to reflect particle break-up and remineralisation by heterotrophic bacteria and/or zooplankton (Buesseler et al., 2008; Maiti et al., 2010; Savoye et al., 2004).

- *Line 495: Similar remineralization although one study was conducted in the tropical Pacific and the other in the North Atlantic Ocean. If the authors want to provide that comparison it might be interesting to discuss a bit the similarities and differences between the studies that lead to comparable values (although some higher values were reported in the tropical Pacific) since one could expect different planktonic communities in both regions, leading to different remineralization intensities.*

Right, thank you. Given the fact that both studied areas (the North Atlantic and the oxygen minimum zone of the tropical Pacific ocean) are very different, we preferred to remove this comparison. However, as Black et al. (2017) were presenting R100 values for the first time (to my knowledge), we kept citing their work when presenting the calculation of the R100 values.

Lines 188-193: To estimate the intensity of shallow remineralization, export flux was also calculated for the Eq+100 m depth horizon. In case of any $^{234}$Th excess below Eq due to remineralisation, export fluxes integrated until Eq+100 m will be less than when integrated until Eq. Following Black et al. (2017) the reduction of the $^{234}$Th flux, R100, is expressed as:

$$R100 = P_{Eq} - P_{Eq+100} \qquad (5)$$

where R100 is the flux reduction in dpm m$^{-2}$ d$^{-1}$ and P is the $^{234}$Th export flux estimated at Eq or Eq+100.

- *Line 509: Stipulate in the following "section".*

OK. This has been checked and modified in the entire manuscript.

- *Line 583: Maybe specify that the extrapolation curves from Fig 3 were used to obtain the deep POC to $^{234}$Th ratios.*

OK.

Lines 534-537: Note that the POC export flux at Eq+100 (Table 3) was calculated by multiplying the $^{234}$Th flux at Eq+100 by the POC to $^{234}$Th ratio of large particles for the same depth. The POC:$^{234}$Th ratio at Eq+100 was deduced from a power law fit (Fig. 3).

- *Lines 642-644 (and previously mentioned too): Could the authors provide a potential cause of that enhanced remineralization in the cold waters of the Labrador Sea, especially since the biogenic Ba$_{xs}$ also shows signs of remineralization. Is it also due to bacterial activity? For how it is written it looks like the authors believe is not due to bacterial activity.*

Right, the potential cause of the enhanced remineralization in the Labrador Sea proposed in Lemaitre et al. (2018) has been added in Section 4.4.

Lines 585-590: This is also in agreement with the highest R100 and carbon remineralisation flux determined with the Ba$_{xs}$ proxy (Lemaitre et al., 2018). The central Labrador basin, in proximity of Station 69, was characterized by strong subduction of the LSW during the winter preceding the GEOVIDE cruise. This downwelling could have promoted an important organic matter export leading to important prokaryotic heterotrophic activity in mesopelagic waters. This enhanced remineralisation was still observed during GEOVIDE as traced by a large mesopelagic Ba$_{xs}$ content (Lemaitre et al., 2018).

- *Line 651: This statement is not strictly quantitatively proven and although the authors provide the date of the peak of the bloom and PP values, they do not refer to the intensity of the bloom (intensity meaning magnitude of PP? Duration of the bloom? Duration of the bloom with sustained high PP values?).*

This is a very good point as the intensity of the bloom can be defined in different ways: at sampling time (illustrated by the *in-situ* PP), as the maximal PP intensity along the season or as an average PP intensity along the season.
In Section 4.2, we propose explanations for the magnitude of the POC exports according to these different definitions. The low seasonal PP average at Station 13 is suggested to explain the low POC export there.

Lines 474-476: One of the lowest POC export flux was determined at Station 13 in the Iberian basin, where the intensity of the bloom remained rather low along the season (seasonal VGPM-PP=81 mmol m$^{-2}$ d$^{-1}$, Fig. 5) due to oligotrophic conditions (depleted nutrients; Fonseca-Batista et al., 2018).

The maximal PP intensity (highest PP peak along the season) observed just before the sampling in the west European basin might explain the high POC export there.

Lines 483-484: PP appeared maximal just before the sampling in the west European basin (Fig. 4 and 5) and could have promoted these high POC export.

The high *in-situ* PP intensity in the Irminger basin indicates that the bloom is reaching its maximum and that export did not yet start at sampling time.

Lines 490-493: Indeed, this area had the highest *in-situ* PP, a high proportion of particulate [234]Th in surface waters (reaching 94% of the total [234]Th activity at Station 44) and a very low P/J ratio, indicating that [234]Th was retained in the upper waters rather than being exported (Fig. 6; Table 1).

> *The authors discuss the temporality of the bloom with respect to the sampling time, which has been done in previous studies, but it could be interesting to produce a figure or correlation between the stage of the bloom (and/or intensity of the bloom, if defined) and the magnitude of the POC to support this statement in a more quantitative manner to be able to say that they are, in fact, directly related.*

Thank you very much for this great suggestion. In Figure 8, we attempt to illustrate the impact of the intensity of the bloom at sampling time (using the *in-situ* PP values), the stage of the bloom (using the percentage of the *in-situ* PP relative to the maximal VGPM-PP along the season) and the different phytoplankton communities on the POC export fluxes. A significant negative correlation is found between the stage of the bloom and the POC export when not considering the stations sampled between two PP peaks (or before the PP peak: Stations 26, 32, 38). In this figure, we can also see that the stations sampled close to the bloom maximum (%max seasonal PP > 80% and with a high *in-situ* PP intensity) are characterized by low POC exports.

[Figure]

Figure 8: Percentage of the *in-situ* primary productivity (PP) relative to the maximal VGPM-PP along the season (%max seasonal primary productivity) in function of the POC export fluxes

determined at the Eq depth. The %max seasonal primary productivity illustrates the stage of the bloom (i.e., a %max seasonal primary productivity equalling 100% corresponds to a sampling time at the bloom peak). This relationship is significant when not taking into account the stations sampled between two PP peaks (Stations 26, 32 and 38, see Fig. 4): $R^2$=0.77 and p-value<0.01. The *in-situ* PP measured at sampling time is indicated with the colours in order to indicate the bloom intensity. The dominating phytoplankton community is also indicated, with circles indicating micro-phytoplankton dominance (with a majority of diatoms), triangles nano-phytoplankton dominance (with a majority of haptophytes) and diamonds pico-phytoplankton dominance (with a majority of cyanobacteria). Note that Station 1 is represented by a star because of the mixed proportion of micro-, nano- and pico-phytoplankton.

- *Line 660: I would delete the first sentence of the point iii) of the conclusions because that is not something that has been studied in this manuscript, it is probably going to be done in the coming Lemaitre et al. in prep. manuscript.*

Right, details on this future paper has been removed, especially the potential impact of the lithogenic particles. However, the potential impact of the particle density related to the presence of diatoms or coccolithophorids on export is still reported.

Lines 598-603: The magnitude of the fluxes seems also to be related to the phytoplankton size and community structure. One of the lowest POC export fluxes was found at the stations where pico-phytoplankton dominated the community. In contrast, the areas composed by micro- and nano-phytoplankton were characterized by high POC export fluxes. These areas were dominated by diatoms or coccolithophorids, known to strongly ballast the POC export fluxes. This suggests that the size as well as the composition and density of the particles likely play an important role on the particulate sinking velocities and thus on the magnitude of the POC export fluxes.

**Anonymous Referee 2**

- *General comment: However the manuscript is missing clear motivations and objectives (see comment 1). This shortcoming has an impact throughout the manuscript, which is tedious to read and not as informative as it could be (see comment #2). The manuscript is long but new results advertised in the abstract are not clearly highlighted and discussed in the main text (e.g. control from phytoplankton size structure and the stage of the bloom). The result section is a very descriptive listing of all measured parameters and the discussion ressembles a result section (comments #3 and #4). I recommend major revisions to improve the readability and strengthen the main points.*

We thank the referee for all the suggestions on how to improve our manuscript. We paid special attention to the definition of our objectives, limiting the redundancy and improving our conclusions. To do so, the structure of the manuscript substantially changed. As you suggested, the Introduction section points out more clearly the question we try to address with this dataset, the Result section is now organized by biogeochemical basins and the sub sections of the Discussion are now based on the different factors influencing the magnitude of the POC exports (stage and intensity of the bloom, phytoplankton size and community structure). We hope that you will be satisfied with our detailed answers below.

- *Major comment 1: The manuscript lacks a clear objective. In Line 61, "According to the impact of these biogeochemical factors [. . .], the efficiency of the NAtl to transfer POC .. can be questioned. In this context, we investigated the . . . export using Thorium." How is the state-of-the-art presented between L32 to 60 questioning the transfert efficiency established in previous studies? After presenting the state-of-the-art, I strongly encourage the authors to present what open question or inconsistency they are trying to address with their dataset. Possible avenues are: What is missing in previous studies? How is this dataset complementary or inconsistent with previous data? I suggest following the traditional structure: 1_ Previous studies showed that X . . .. 2_ However, Y is still unknown (or this is inconsistent with Z); 3_ Here, we examine/show/leverage. . .. This objective should also guide the reader in the result and discussion section (see comment 2).*

Thank you for your great help, this comment is useful for this manuscript but will also be useful for the next ones. As suggested, the introduction section has been re written in order to clarify the importance of our study: 1) Carbon export in the North Atlantic has been well studied but, a substantial range of carbon export efficiencies has been reported by earlier studies at different locations of the North Atlantic. This is directly questioning about how carbon export efficiency varies at a trans-Atlantic scale and what are the controlling factors 2) The North Atlantic is characterized by different biogeochemical basins, defined by different trophic states, phytoplankton communities and hydrodynamic processes These distinct biogeochemical factors impact strongly the POC export magnitude. 3) Therefore, we examine here the impact of the stage and intensity of the bloom and the phytoplankton structure on POC export fluxes, before to evaluate the export and transfer efficiency of the high latitude North Atlantic basin.

[revised manuscript text omitted]

- *Major comment 2: The result section appears as a long list of parameters (e.g. 3.3- Particulate Th and POC distribution, 3.4- POC:Th ratios, 3.5-POC export; 3.6- PP), and include too many methodological details. For example: L 287 to 292 "LSF particles are collected on silver GF/F filters . . ."; L329-332; L377 "Using the 8-day average data, PP was estimated for the preceding month and the whole productive period. . . ". The result section should be re-worked to emphasize the important connections between the different measured parameters (export, PP, planktonic composition etc.). One option would be to present the results per biogeochemical province and make these links. Another option would be to organise the result section based on processes and/or novel findings (for example: Control by phytoplankton size structure, Modulation by stage of the bloom . . . or Flux attenuation in mesopelagic zone, which is now in the discussion but would fit better in as a result section- see comment #4). Please move methodological details to method section or remove when it is duplicated.*

Thank you for your help. As suggested, the Result section has been re written to describe each biogeochemical basin individually. Many details have been deleted or moved to the Method section.

[revised manuscript text omitted]

- *Major comment 3: The discussion of uncertainty opens the discussion (sections 4.1 and 4.2 NSS and physical transport). While I value this discussion, it is not novel and has been discussed in previous studies. I suggest to move it to the end of the manuscript. Please start with what is new and motivating before discussing the limitations of the method.*

It is a very good point. However, we think that it is easier for the reader to know about the potential bias of the $^{234}$Th method before reading the sections explaining the flux estimates. If the potential limitations were discussed at the end of the manuscript, the reader would have to re consider the explanations that have been previously stated. We thus decided to keep this section at the beginning of the Discussion, as it is the case in other papers about the POC exports using by the $^{234}$Th-based approach (e.g., Black et al., 2017; Owens et al., 2015). We nevertheless decided to group both sections in one named "Validity of the export estimations", which is hopefully more motivating for the reader.

- *Major comment 4: Large part of the discussion pertains to the result section (sections 4.3, 4.4 and 4.5) and could help organise the results (see comment #2). There are also many methodological details in the discussion section that should be (re)moved (e.g. L446, L462, L483), in particular when these details are stated several times. For example, L483 explains what thorium deficit is, even though this explanation is already included in the introduction, the method and the result sections. The discussion should emphasize what this study brings to existing studies and discuss the limitations.*

Thank you for noticing all these methodological details. Many parts of the discussion have been moved to the Introduction, Method or Result sections. We also decided to re organize the discussion section based on the different processes influencing the magnitude of the POC exports, as suggested in your major comment n°2.

[revised manuscript text omitted]

- *Major comment 5: please streamline the text. Many methodological concepts are presented and introduced in several sections (e.d. thorium deficit, PP measurements etc.).*

OK, many parts along the manuscript have been deleted or moved to the right section, and we hope the new text will be less tedious to read.

- *Lines 43-46: two sentences are repeated in the introduction. Remove one version.*

The introduction has been re written and this repeated sentence has been removed.

- *Lines 61-63: needs to be rephrased (see comments above).*

OK. This has been modified, please see our answer to your major comment n°1.

- *Method section: could be sharpened by limiting the use of "moderate" (e.g. L91, L106, L116). Please limit the use of "moderate" (e.g. L91, L106, L116), which is rather vague. The word "briefly" (L127, L137, L222) should also be avoided. Either you have described the method and you can remove "briefly", or you haven't described enough and should include additional references or details.*

Right, many details of the Method section have been deleted and the use of words such as "briefly" have been removed. When using the word "moderate", result values have been added in the text in order to give the order of magnitude.

Line 260: …. and a moderate PP at Station 13 (79 mmol m$^{-2}$ d$^{-1}$; Fonseca-Batista et al., 2018; this issue)

Line 341: POC:$^{234}$Th ratios were moderate for both size fractions, reaching 14 µmol dpm$^{-1}$ for the SSF at Station 44…

Lines 370-371: POC concentrations and particulate $^{234}$Th activities were moderate to low, except at Station 77 where values were higher in the surface, reaching 11 µmol L$^{-1}$ and 0.45 dpm L$^{-1}$ for the SSF…

Lines 381-382: ... but, in general, the $^{234}$Th export fluxes of the Labrador basin were moderate, averaging 758 dpm m$^{-2}$ d$^{-1}$ (Table 1).

- *Line 102: associated with (not to).*

OK, this has been changed.

- *Line 235: other estimates of PP mentioned here could be added for comparison in Figure 4. This would give confidence in the author's choice and inform the reader on the uncertainty associated with these estimates.*

OK. We added the PP estimates from the Eppley-VGPM and CbPM models in Figure 4.

[Figure]

Figure 3: *In-situ* (squares) and satellite VGPM-derived (continuous lines), VGPM-Eppley-derived (dotted lines) and CbPM-derived (dashed lines) primary production (PP; in mmol m$^{-2}$ d$^{-1}$) data at the time of our sampling and along the year 2014. The start of the bloom, defined by a PP increase of 30% above the winter value, is indicated with the black vertical dashed line.

- *Lines 259-261: Move definition of PPZ above, when it is first mentioned.*

OK. This has been done.

Lines 169-172: Eq. 3 has been solved for z taken as the depth (Eq) at the base of the $^{234}$Th deficit zone (Eq = depth where $^{234}$Th activity is back to secular equilibrium with $^{238}$U) as well as for z representing the base of the primary production zone (PPZ), i.e. the depth where in-situ fluorescence was only 10% of its maximum value (Owens et al., 2014).

- *Line 110: Tonnard et al, in prep and L521 Lemaitre et al, in prep. Does the journal authorized unpublished papers?*

The citation "Lemaitre et al., in prep" and thus the details related to this manuscript have been removed. The manuscript "Tonnard et al., in prep." is going to be submitted soon, hopefully before the publication of this manuscript.

*General comments: You have provided detailed replies to the reports and taken into account all the major comments received. In particular, you have considerably streamlined the paper to clarify objectives, provided more clearly the context and discussed the major results. Thus, I don't think that there is room for a second round of major revision. I have read the manuscript and have added a few comments or questions. I'd like you to take them into account and post a revised manuscript.*

We would like to thank the Editor for these very positive comments. All suggestions have been taken into account and are detailed below.

- *Line 190: Number missing*

OK, this has been changed.

Line 189: In case of any $^{234}$Th excess relative to $^{238}$U (i.e., $^{234}$Th/$^{238}$U ratio > 1) below the Eq depth…

- *Line 195: Instead of "removal", maybe use "transfer"*

OK.

Line 195: To estimate the transfer rate of $^{234}$Th from the dissolved to the particulate form…

- *Line 218: root sum of square method or root sum squared method ?*

Line 218: …using a root sum squared method…

- *Line 239: 'at few stations'? do you mean that 42 m happened at more than one station? (or is it for a specific station)?*

Lines 238-240: The 0.2% of surface PAR depth was roughly corresponding to the Eq depth (median difference between both depths: 20 ± 13 m) although, a 42 m difference was observed at Station 1.

- *Line 268: a ratio of 28%...*

Lines 267-269: Interestingly, these two stations vary also in their total particulate $^{234}$Th (sum of the SFF and LSF) over total $^{234}$Th ratios with only 9% of the $^{234}$Th in the particulate phase at Station 1 and a ratio of 28% at Station 13 (in the median of those observed elsewhere along the transect).

- *Line 279: albeit with a strong associated error..*

Okay, that has been changed (see line 280).

- *Line 283: During 2014, the ... or : before the cruise in 2014,...*

Line 284: During 2014, the west European basin..

- *Line 285: Altogether, sampling coincided with the bloom development in this basin, with in-situ PP*

Lines 286-287: Altogether, sampling coincided with the bloom development in this basin, with *in-situ* PP reaching 135 and 174 mmol m$^{-2}$ d$^{-1}$ at Stations 21 and 26, respectively (Table 3).

- *Line 287: Remove "the"*

Line 288: Along with high PP..

- *Line 299: of nearly 0.5 for both stations indicates...*

Lines 300-301: The resulting export ratio (P/J) was close to 0.5 for both stations, indicating a balanced situation between export and scavenging fluxes.

- *Line 302: was significantly positive*

Okay, see line 303.

- *Line 310: presented similar characteristics to those...*

Line 311-312:  In general, the different fluxes in the Icelandic basin presented similar characteristics to those in the west European basin.

- *Line 314: but unlike for the west European basin*

Okay (see line 316).

- *Line 317: between the two size fractions*

Okay (see line 319)

- *Line 330: are lower than most values... (just after you mention values as low as 0.8, which are smaller?)*

The study of Martin et al. (2011) report a POC export flux of 0.8 mmol m$^{-2}$ d$^{-1}$ indeed, but this is an extreme minimum compared to the fluxes generally observed in this basin.

Lines 332-334: Such POC export fluxes are lower than most values reported in earlier studies, ranging from 6 to up to 52 mmol m$^{-2}$ d$^{-1}$ although Martin et al. (2011) determined a very low value of 0.8 mmol m$^{-2}$ d$^{-1}$.

- *Line 351: "On the other hand, this high..." I wondered whether this was not an issue of non-stationarity at station 44.*

Right, and the non-stationarity of Station 44, and more generally of the Irminger basin, is discussed in different sections:

Line 418: The Irminger basin in spring is a really patchy and dynamic area..

Lines 494-495: The Irminger basin was sampled close to the bloom maximum, but unlike the west European and Icelandic basins the POC export flux was low there, probably reflecting accumulation of biomass preceding export.

Lines 354: On the other hand, this high particulate fraction in the upper layer did not induce a high export flux.

- *Line 363: for the month before and after...*

Okay, see line 366.

- *Line 374: ratios were similar in the two size fractions,*

Okay, see lines 376-377.

- *Line 397: with respect to*

Okay, see line 400.

- *Line 411: under the influence*

Okay, see line 414.

- *Line 412: the impact of lateral advection*

Okay, see line 415

- *Line 422: Hydrodynamic processes could also impact open ocean sites, such as...*

Okay, see line 425.

- *Line 436: Question: and the fact that convection has ended probably more than 2 (if not 3) months ahead of the cruise, thus much longer than half life... would also be an argument against invoking it...*

That is completely correct and this has been added.

Lines 439-440: Moreover, this convection ended more than two months before our sampling, a time lag that largely exceeds the [234]Th half-life thereby erasing any potential impact on the [234]Th signal.

- *Line 437: this is not 'vertical molecular diffusion' that you estimate, but possible 'vertical mixing/diffusion'...*

Okay that has been changed (see line 442).

- *Line 450: I don't see why it should systematically lead to an underestimate... (as it would depend whether one samples after bloom peak or before)*

Right, the use of a SS model during the North Atlantic spring bloom (characterized by a strong PP variability) leads to a less accurate estimate of the export but not necessarily results in underestimating it.

Lines 454-456: As a consequence, the SS model was shown to poorly describe the magnitude of the [234]Th export flux, leading to differences with the NSS model up to a factor of 3 (Buesseler et al., 1992; Martin et al., 2011).

- *Line 552: has also been observed...*

Okay, see line 557.

- *Line 560: replace 'the period of our study' by 'late spring' (or something else)*

Okay, see line 565.

- *Line 564: Strong variability of PP during this longer period would strongly impact the ratio.*

Lines 568-569: Strong variability of PP during this longer period would highly impact the ThE ratio.

- Line 579: could it also reflect lower particle sinking speeds?

Indeed, thank you.

Lines 581-585: The low T100 (and high R100) values observed in the eastern part of the transect (Stations 1, 13 and, to a lesser extent Station 21) likely reflect an important bacterial activity in these warmer waters (>13°C in the upper 100 m; Iversen and Ploug, 2013; Marsay et al., 2015; Rivkin and Legendre, 2001), efficiently degrading the probably slow-sinking particles. Such recycling is characteristic for regeneration-based microbial food webs in oligotrophic regimes (Karl, 1999; Thomalla et al., 2006).

- *Line 594: over a month duration or during the 1-month long cruise*

Okay, see line 599.

- *Line 594: large temporal variability*

Okay, see line 599.

- *Line 598: seems also related...*

Okay, see line 603.

- *the black letters are hard to read on dark (blue or purple) circles. (maybe change them to white...)*

This is right, and we changed the colour of the numbers.

[Figure]

- *remove 'determined'*

Okay, see Figure 8.

- *Replaced 'determined' by 'measured on'*

Okay, see Figure S1.